# Emergence of flat bands and ferromagnetic fluctuations via orbital-selective electron correlations in Mn-based kagome metal

Subhasis Samanta [1,2], Hwiwoo Park[3], Chanhyeon Lee[4], Sungmin Jeon[3], Hengbo Cui[5], Yong-Xin Yao [6,7], Jungseek Hwang [3] ✉, Kwang-Yong Choi [3] ✉ & Heung-Sik Kim [1] ✉

Kagome lattice has been actively studied for the possible realization of frustration-induced two-dimensional flat bands and a number of correlation-induced phases. Currently, the search for kagome systems with a nearly dispersionless flat band close to the Fermi level is ongoing. Here, by combining theoretical and experimental tools, we present $Sc_3Mn_3Al_7Si_5$ as a novel realization of correlation-induced almost-flat bands in the kagome lattice in the vicinity of the Fermi level. Our magnetic susceptibility, $^{27}Al$ nuclear magnetic resonance, transport, and optical conductivity measurements provide signatures of a correlated metallic phase with tantalizing ferromagnetic instability. Our dynamical mean-field calculations suggest that such ferromagnetic instability observed originates from the formation of nearly flat dispersions close to the Fermi level, where electron correlations induce strong orbital-selective renormalization and manifestation of the kagome-frustrated bands. In addition, a significant negative magnetoresistance signal is observed, which can be attributed to the suppression of flat-band-induced ferromagnetic fluctuation, which further supports the formation of flat bands in this compound. These findings broaden a new prospect to harness correlated topological phases via multiorbital correlations in $3d$-based kagome systems.

A flat band system is characterized by the presence of a dispersionless energy band, where the group velocity of electrons vanishes at every crystal momentum[1,2]. Because quenching of kinetic energy promotes the effects of electron correlations, flat band systems offer an ideal platform to examine strongly correlated quantum phenomena, encompassing fractional Chern insulator phases and unconventional superconductivity[3–6]. Theoretically, flat bands have been studied in dice, Lieb, kagome, honeycomb, and Tasaki's decorated square lattices, where destructive quantum interference from two or more hopping channels produces a flat band[7–11]. Experimentally, localized flat band state has been reported in photonic Lieb and kagome lattices[12–14] as well as in realistic condensed matter systems[15–17].

Among the family of flat band systems, the kagome lattice has been one of the most studied. Geometric frustration inherent in the kagome lattice creates destructive interferences between multiple nearest-neighbor electron hopping channels and yields flat bands[17,18].

[1]Department of Semiconductor Physics and Institute of Quantum Convergence Technology, Kangwon National University, Chuncheon 24341, Republic of Korea. [2]Center for Extreme Quantum Matter and Functionality, Sungkyunkwan University, Suwon 16419, Republic of Korea. [3]Department of Physics, Sungkyunkwan University, Suwon 16419, Republic of Korea. [4]Department of Physics, Chung-Ang University, Seoul 06974, Republic of Korea. [5]Department of Physics and Astronomy and Institute of Applied Physics, Seoul National University, Seoul 151-747, Republic of Korea. [6]Ames National Laboratory, U.S. Department of Energy, Ames, IA 50011, USA. [7]Department of Physics and Astronomy, Iowa State University, Ames, IA 50011, USA. ✉e-mail: jungseek@skku.edu; choisky99@skku.edu; heungsikim@kangwon.ac.kr

Because of zero kinetic energy within ideal flat bands, the impacts of electron correlations with such bands can be maximized. Experimentally, engineering flat band systems enables a viable route to realize correlation-induced emergent phenomena such as Chern insulators, fractional quantum Hall states, quantum spin liquid, superconductivity, and topological magnon insulators[19–25]. Nonetheless, the realization of ideal flat dispersions has remained elusive due to the presence of long-range hopping paths and the difficulties in placing flat bands in the vicinity of the Fermi level ($E_F$).

Recently, 3d-transition metal-based two-dimensional layered kagome compounds have captured much research attention because of their multiorbital nature. The most studied compounds are $Co_3Sn_2S_2$[26–28], $Fe_3Sn_2$[29–31], $FeSn$[32], $CoSn$[17,33,34], $YCr_6Ge_6$[35], and $YMn_6Sn_6$[36]. Thanks to the active 3d-orbital degree of freedom in all the compounds listed above, multiorbital phenomena such as orbital magnetization or spin-orbit coupling (SOC)-induced topological properties and anomalous (spin-)Hall responses have been actively explored.

At this point, an interesting question arises about the role of electron correlations in these multiorbital kagome systems, where kagome-induced flat bands coexist with other dispersive bands that are less affected by the kagome-induced destructive interference. In non-kagome multiorbital compounds such as $Ca_{2-x}Sr_xRuO_4$[37] and Fe-based superconductors[38–41], nontrivial orbital-dependent correlation effects induced by the on-site Coulomb repulsions and Hund's coupling have been reported, such as orbital-dependent Mott transitions[42–46] and Hund's metallic phases[47–50]. However, the impact of electron correlations on the electronic structure, especially in the presence of kagome-induced flat bands and SOC in realistic systems, has not been much discussed in previous studies. (It was speculated in ref. 36 that $YMn_6Sn_6$ can be a Hund's metal, but without further elaboration.)

In this work, we study the electronic and magnetic properties of Mn-based kagome metal $Sc_3Mn_3Al_7Si_5$ (SMAS). This compound crystallizes in a hexagonal structure with a space group $P6_3/mmc$. Figure 1a, b present the crystal structure of SMAS and the underlying Mn kagome network (Fig. 1c showing five Mn d-orbitals in the Mn kagome network schematically). Previous experimental reports on SMAS reveal a predominant metallic character with no signature of static magnetic order down to 1.8 K. The specific heat capacity measurement shows a large Sommerfeld coefficient, suggesting a vital role of electronic correlations[51]. The absence of long-range magnetic order at very low temperatures indicates a strong magnetic fluctuation in this system, as further probed by inelastic neutron scattering measurements[52]. On the other hand, the previously reported magnitude of magnetic moment (0.5 $\mu_B$/Mn), compared to the one from Hund's rule applied to the Mn $d^5$ charge state (S = 5/2), implies an itinerant character of magnetism.

Here, we combine experimental and first-principles calculation tools to explore potential correlation-induced flat-band physics in SMAS. Our magnetization, magnetic susceptibility, and $^{27}Al$ nuclear magnetic resonance (NMR) measurements indicate the presence of ferromagnetic fluctuations below T < 30 K, which is attributed to the formation of flat bands and potential negative magnetoresistance in flat band systems[53–55]. From ab-initio density functional and dynamical mean-field calculations we find that correlation-induced flat bands emerge in the vicinity of the $E_F$ at $k_z = 0$, which are likely to be strongly linked to the ferromagnetic fluctuations observed in the low-T regime. We propose that the flat bands are induced by i) kagome-induced geometric frustration within a subset of Mn d-orbitals, revealed by constructing electronic Wannier orbitals from the non-correlated electronic structure as depicted in Fig. 1c, and ii) orbital-selective electron correlations which selectively push the kagome-induced weakly dispersive bands up to the $E_F$ and strongly renormalize the bandwidth (see Fig. 1d for a schematic illustration). We further observed a significant negative magnetoresistance in this system, which supports the presence of flat-band-induced ferromagnetic fluctuations in SMAS as suggested in $CoSn$[56]. Additionally, our

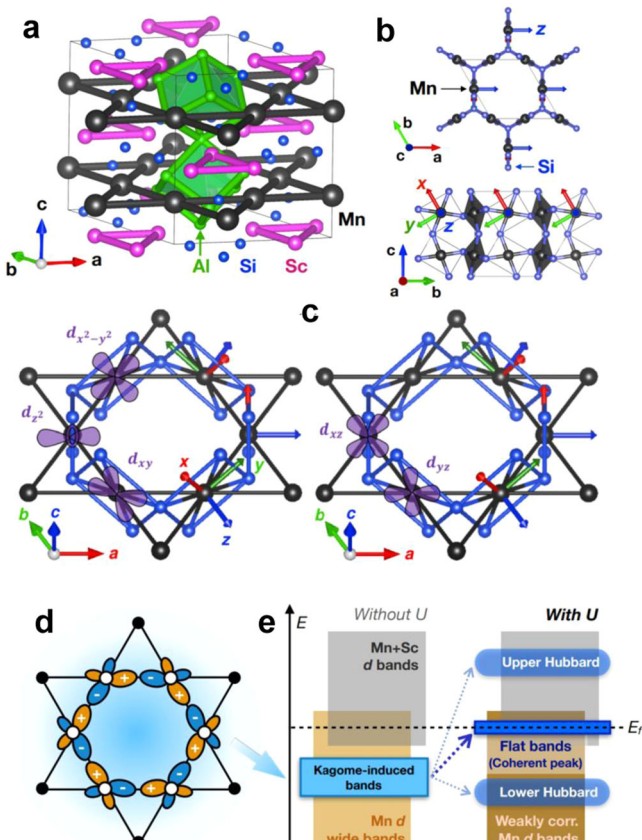

**Fig. 1 | Overview of crystal structure and correlation-induced kagome flat bands. a** Crystal structure of SMAS highlights the formation of a Mn kagome network, Sc equilateral triangles, and distorted $Al_8$ cubes. **b** Top and side views of the structure show the connectivity between Mn and Si atoms and the formation of $MnSi_4$ rectangles, constituting a three-dimensional network. Note that the local cartesian axes for the definition of Mn d-orbitals are depicted as red, green, and blue arrows. **c** Schematic shape and orientation of Mn d-orbitals. **d, e** Schematic illustrations of Wannier orbital realizing kagome-induced weakly dispersive bands in SMAS and shift of the kagome-induced bands up to the $E_F$ (in addition to the formation of lower- and upper-Hubbard bands) via orbital-selective electron correlations.

dynamical mean-field calculations show a slight upturn in DC resistivity in the low-temperature regime, consistent with our DC resistivity measurement, which can be attributed to the enhanced orbital susceptibility and ferromagnetic fluctuations. These findings make SMAS a promising platform for further exploring correlated and topological phenomena emerging from flat band systems.

## Results

### Experimental signatures of ferromagnetic instabilities

Figure 2 a presents the temperature and field dependencies of the electrical resistivity $\rho(T)$ of SMAS. On cooling, $\rho(T)$ decreases down to 30 K and reaches a minimum at 25 K, below which it experiences an increase. We note that the essentially same transport behavior was observed in the previous study[51]. As evident in the inset of Fig. 2a, the application of an external magnetic field somewhat suppresses $\rho(T)$. This observed upturn below 25 K alludes to the development of an additional scattering channel.

Figure 2 b shows the temperature dependence of the in-plane and out-of-plane magnetic susceptibilities $\chi(T)$ measured in an applied field of 0.1 T. With decreasing temperature, $\chi(T)$ increases steeply with no indication of saturation or anomaly, thereby excluding the occurrence of long-range magnetic ordering. Upon closer inspection of $\chi(T)$, we

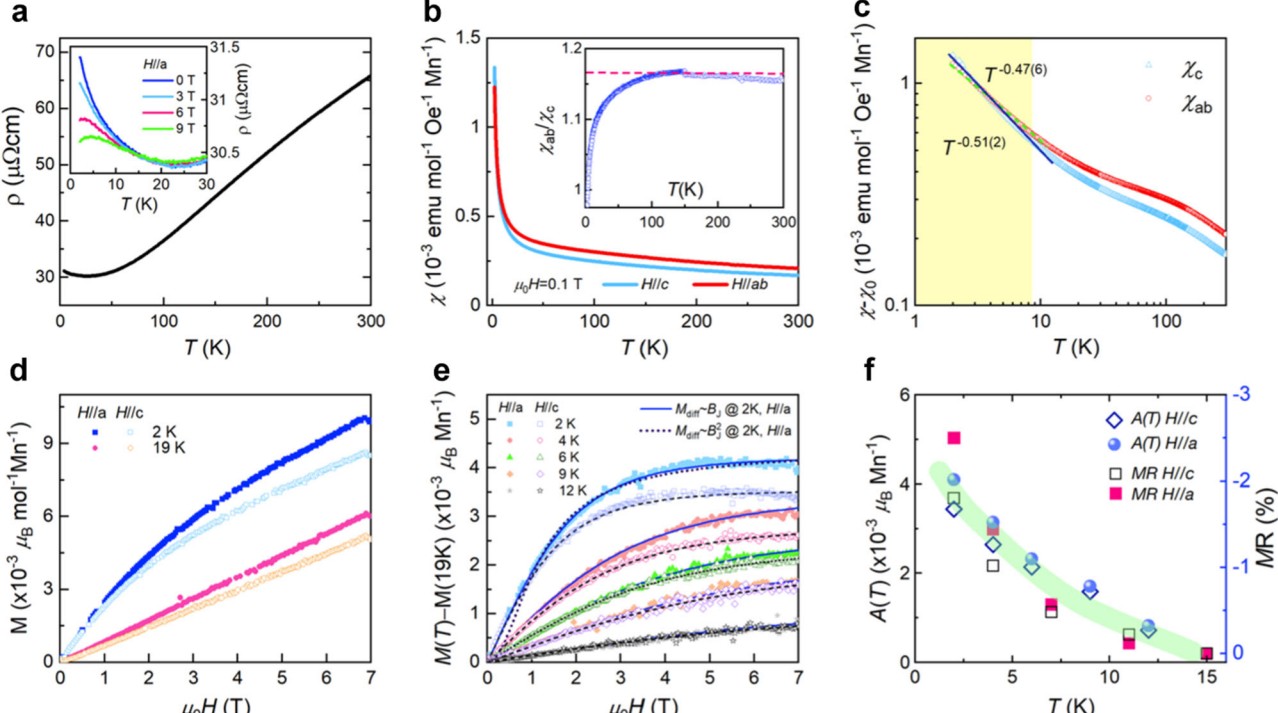

**Fig. 2 | Electrical and magnetic properties of SMAS. a** Temperature-dependent electrical resistivity $\rho(T)$ of SMAS. The inset plots the field-dependent $\rho(T, H)$ for $H//a$. **b** Temperature dependence of the static magnetic susceptibility $\chi(T)$ measured at $\mu_0 H = 0.1$ T for $H//ab$ and $H//c$, with an inset showing the ratio of in-plane $\chi_{ab}(T)$ to out-of-plane $\chi_c(T)$. **c** Log-log plot of $\chi_{ab} - \chi_0$ (red circles) and $\chi_c - \chi_0$ (cyan triangles) versus temperature. The solid and dashed lines represent fits to a power-law dependence $\chi(T) \cdot T^{-\alpha}$ at low temperatures. **d** Magnetization curves $M(H, T)$ at $T = 2$ K and 19 K for $H//c$ (open symbols) and $H//a$ (full symbols). **e** Difference of the magnetization curves $M(H, T) - M(H, T = 19$ K$)$ at selected temperatures $T = 2, 4, 6, 9,$ and 12 K with solid and dashed lines indicating fits to a modified Brillouin function as described in the text. Note that an additional $M_{\text{diff}} \sim B_J^2$ fitting for $H//a$ at $T = 2$ K, where $B_J$ is the Brillouin function as mentioned in the text, is depicted as a dotted line. **f** Temperature dependence of the amplitude parameter $A(T)$ for $H//c$ (diamonds) and $H//a$ (spheres) and magnetoresistance at $B = 9$ T for $H//c$ (open squares) and $H//a$ (full squares) as a function of temperature. The thick line is a guide to the eye.

observe a notable disparity between the in-plane $\chi_{ab}(T)$ and the out-of-plane $\chi_c(T)$. To quantitatively assess the temperature-dependent magnetic anisotropy, we plot the ratio $\chi_{ab}(T)/\chi_c(T)$ in the inset of Fig. 2b. Remarkably, a broad maximum is observed with $\chi_{ab}/\chi_c \approx 1.16$ at approximately $T \sim 145$ K. The decrease in $\chi_{ab}/\chi_c$ below 145 K implies that a weak XY-like magnetism becomes increasingly isotropic as $T \to 0$ K.

To elucidate the anomalous behaviors of the magnetic susceptibility, we first estimate the constant contribution to $\chi(T)$, $\chi_0 = \mu_0 N_A \mu_B^2 D(\epsilon_F) = 7.8 \times 10^{-5}$ emu $\cdot$ Oe$^{-1} \cdot$ mol$^{-1} \cdot$ Mn$^{-1}$, where $N_A$ is Avogadro's number and $\mu_B$ is the Bohr magneton. This value is obtained from our DFT calculations, where $D(\epsilon_F) = 7.24$ states/eV/formula unit represents the density of states at the Fermi level. In Fig. 2c, the $\chi_0$-subtracted magnetic susceptibilities $\chi_{ab}(T) - \chi_0$ and $\chi_c(T) - \chi_0$ are displayed on a log-log scale. Notable changes in slope and anisotropy are observed around 130 K, where the maximum ratio $\chi_{ab}(T)/\chi_c(T)$ occurs, and between 10 K and 25 K, coinciding with the resistivity minimum. The multi-stage evolution of anisotropic magnetic correlations points to the presence of multiple underlying energy scales. Below 8 K, a power-law increase becomes apparent with $(\chi_{ab} - \chi_0)(T) \sim T^{-0.47(6)}$ and $(\chi_c - \chi_0)(T) \sim T^{-0.51(2)}$, signifying the development of critical-like ferromagnetic correlations. Further Curie-Weiss (CW) analysis of $1/(\chi_{ab}(T) - \chi_0)$ above 150 K yields the effective magnetic moment of $\mu_{\text{eff}}^{ab} = 0.86(3)$ $\mu_B$/Mn and the CW temperature $\theta_{\text{CW}}^{ab} = -421.(4)$ K, and from $1/(\chi_c(T) - \chi_0)$ (above 150 K) we obtain $\mu_{\text{eff}}^c = 0.87(2)$ $\mu_B$/Mn and the CW temperature $\theta_{\text{CW}}^c = -368.(9)$ K. The significantly reduced effective moment, compared to the spin-only value of 4.97 $\mu_B$ expected for Mn$^{3+}$ ions, suggests a dominant itinerant character of the magnetism. These CW parameters are, thus, regarded as indicators of correlation-driven magnetism.

Isothermal magnetization curves $M(H, T)$ at temperatures $T = 2$ and 19 K for $H//c$ (open symbols) and $H//a$ (full symbols) are shown in Fig. 2d. At 19 K, $M(H)$ exhibits a linear increase, characteristic of a paramagnetic-like state. As the temperature decreases below 19 K, $M(H)$ progressively develops a convex curvature, indicating the emergence of ferromagnetic correlations. This behavior is consistent with the observed upturn in $\rho(T)$ below 25 K and the power-law increase in $\chi(T)$. To further assess the ferromagnetic correlations, we subtract the linear term from $M(H, T)$ and plot the resulting difference in magnetization curves $M_{\text{diff}} = M(H, T) - M(H, T = 19$ K$)$, as shown in Fig. 2e. We attempted to model $M_{\text{diff}}(H, T)$ using a modified Brillouin function $B_J$, defined as $M_{\text{diff}} = A(T)B_J(g\mu_B J(T)B/k_B T)$. Here, $A(T)$ is a temperature-dependent amplitude parameter associated with the saturation magnetization of ferromagnetically correlated spins. With lowering the temperature, the spin moment $J(T)$ may be enhanced due to the orbital-selective amplification of ferromagnetic correlations, yet it hardly varies with an external field. We find that $M_{\text{diff}}(H, T)$ follows a linear Brillouin scaling rather than a quadratic relationship, as demonstrated by comparing the solid line for $M_{\text{diff}}(H//a, T = 2K) \propto B_J$ with the dotted line for $M_{\text{diff}}(H//a, T = 2K) \propto B_J^2$ in Fig. 2e. Noticeably, a similar linear scaling of $M(H) \propto B_J$ is observed in the ferromagnetically ordered state of manganites, which exhibit negative magnetoresistance[57]. In this light, the observed linear Brillouin scaling indicates that the studied system is on the verge of ferromagnetic instability. In Fig. 2f, the extracted values of $A(T)$ are plotted together with MR. Here, $A(T)$ essentially conveys the same information as $J(T)$. The similar trend observed between these parameters establishes a direct relationship between the increasing amplitude of ferromagnetic correlations and the negative MR, thereby supporting the proportionality $M(H) \propto B_J \propto$ MR.

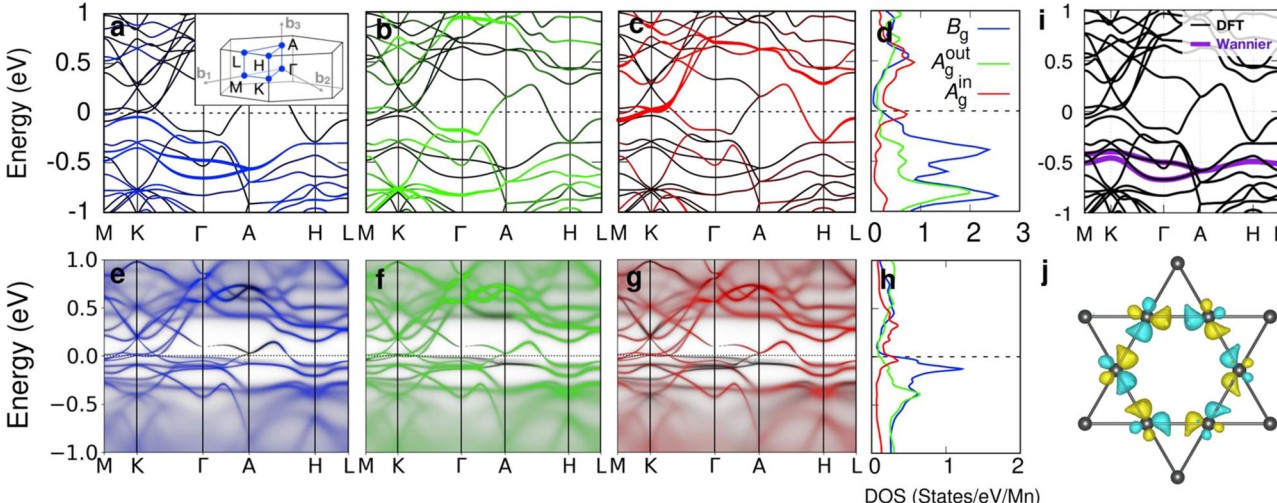

**Fig. 3 | Emergence of correlation-induced flat bands. a–c** Non-spin-polarized band structure of SMAS displays the orbital contribution of Mn $B_g$ (blue), $A_g^{out}$ (green), and $A_g^{in}$ (red) orbitals along the high symmetry paths M-K-Γ-A-H-L, and **d** PDOS obtained using LDA functional in DFT. **e–g** Corresponding orbitally-resolved momentum- and frequency-dependent spectra and, **h** momentum-integrated spectral function from LDA+DMFT calculation for $(U, J_H)$=(6, 0.8) eV at 210 K with full Coulomb interaction. **i** Full DFT bands (black) and Wannier inter-polated $B_g$-bands (violet). **j** Orbital texture with $B_g$ character from constructed Wannier molecular orbitals of non-correlated electronic structure. The inset in **a** depicts the hexagonal Brillouin zone and spacial k-points therein.

The presence of ferromagnetic fluctuations in SMAS is also revealed by our $^{27}$Al NMR results (see Supplementary Note 4). Further, from the NMR result it is also hinted that the resistivity upturn around 30 K (see Fig. 2a) is related to the onset of the ferromagnetic fluctuations and the resulting enhancement of electron scattering. On the other hand, the formation of Mn $d^5$ local moments in SMAS is strongly inhibited due to the strong hybridization between Mn and Si ions. Simple density functional theory calculations (see Fig. 3a–c) or their augmentation with the on-site Coulomb repulsion in a mean-field approximation (*i.e.* DFT+$U$ methods) do not admit ferromagnetism as well (see Supplementary Note 5). Hence, the observed tendency toward ferromagnetic instability becomes a mystery within the standard band-theoretical and mean-field picture. To explain the observed ferromagnetic tendency, the presence of flat dispersions near the $E_F$, induced by the dynamical nature of electron correlations, is required, as discussed in the following sections.

## Flat band realized via correlation-induced orbital-dependent band energy renormalization

For a deeper understanding of the ferromagnetic fluctuations in SMAS as revealed through our experimental measurements, we carried out ab initio electronic structure calculations using DMFT methods. We uncover several peculiar features in this compound: (i) Correlation- and frustration-induced nearly flat band in proximity to $E_F$ at the $k_z = 0$ plane, identified as a correlated metal incipient to an orbital-selective Mott phase. The role of electron correlations shown in Fig. 1d, which selectively promotes kagome bands, is reminiscent of the formation of coherent peak and lower/upper Hubbard bands in the orbital-selective Mott transitions observed in several multiorbital systems such as Ca$_{2−x}$Sr$_x$RuO$_4$ and Fe-based superconductors[38–41]. Hence, the flat bands in SMAS arise from unusual cooperation between kagome-induced kinetic energy quenching and orbital-selective electron correlations, which are likely to be strongly linked to the ferromagnetic fluctuations observed in the low-$T$ regime[58–60], (ii) symmetry-protected metallic nodal surface bands at the $k_z = \pi$ plane, and (iii) strong incoherence driven by Hund's coupling. In this section, we begin with discussing the electronic structure obtained from nonmagnetic DFT calculations.

The top panels of Fig. 3a–c delineate the non-spin-polarized band structure (black lines) calculated using LDA functional without incorporation of additional on-site Coulomb repulsion and spin-orbit

coupling. The colored fat bands in Fig. 3a–c highlight the orbital contributions of Mn $d$ orbitals and the corresponding projected density of states (PDOS) (see Fig. 3d). Note that the Mn $d$ orbitals are labeled in terms of irreducible representations defined with respect to local coordinate axes, where the $z$-axis is set to be parallel to the twofold axis of the $C_{2h}$ point group of the Mn site, as shown in Fig. 1b. The five $d$ orbitals, as depicted in Fig. 1c, can be divided into three groups: $B_g$ (blue) for {$d_{xz}, d_{yz}$}, $A_g^{out}$ (green) for {$d_{x^2−y^2}, d_{xy}$}, and $A_g^{in}$ (red) for $d_{z^2}$. In Fig. 3 and for the rest of the paper, we fix this color coding for Mn $d$ orbitals. We comment that despite all the orbitals being nondegenerate, orbitals belonging to the same irreducible representation show almost identical features in the band structure as well as spectral function. In the vicinity of Γ-point, the topmost occupied band shows a strong $A_g^{out}$-character, which is located higher in energy than the blue $B_g$-bands. Also, note that the bands at the $k_z = \pi$ plane, especially the ones at the $E_F$, are degenerate due to non-symmorphic two-fold screw rotation and time-reversal symmetries in the absence of SOC, consisting of nodal surface bands[61].

The bottom panels of Fig. 3e–g and Fig. 3h show the orbital-resolved spectra and PDOS at 210 K from LDA+DMFT results. In the results presented in Fig. 3e–g, we employed on-site Coulomb parameters $(U, J_H)$ = (6, 0.8) eV. The details on the choice of the parameters and how the result depends on the evolution of parameters will be discussed in the following sections. The quintessential feature of the spectral function is the presence of a nearly flat band, lying just below $E_F$ at the $k_z = 0$ plane, induced by dynamical correlation effect and mostly consisting of $B_g$-orbitals as also clearly shown in the PDOS in Fig. 3h. In the DFT bands, the $B_g$-bands are located around $−0.5$ eV (refer to Fig. 3a, d). In the DMFT results, the $B_g$-bands are pushed up to the $E_F$ with the bandwidth strongly renormalized (see Fig. 3a, e for comparison), while the positions of other bands, especially $A_g^{out}$-bands near Γ and the nodal surface bands at the $k_z = \pi$ plane, remain only weakly affected. The PDOS of $B_g$ flat bands show an emergence of a sharp peak at $E_F$ (compare Fig. 3d, h), which is identified as a coherent peak emerging from correlated metallic phases in the vicinity of Mott transitions[62,63]. This $B_g$-orbital-selective Mott-like correlations will be discussed later in the next section.

To check whether the inclusion of the $U$ parameter on Mn $d$ orbitals plays a similar role in mean-field treatments of the Coulomb repulsion, we performed DFT+$U$ calculations to compute PDOS (see

Supplementary Note 5) and compared them with Fig. 3. Surprisingly, we observe that applying $U$ on Mn $d$-orbitals i) pushes the $B_g$-bands downward, contrary to DFT+DMFT, as shown in Supplementary Note 5 and Supplementary Fig. 4 therein, and that ii) DFT+$U$ favors anti-ferromagnetic order up to $U_{eff}$ = 4 eV, and beyond that $U_{eff}$ value large Mn magnetic moments ($\geq 3\mu_B$) set in. Hence, the renormalization of the $B_g$-bands and the origin of the observed ferromagnetic instabilities with small local moments cannot be captured via the simple DFT or DFT+$U$ description.

As the DMFT spectral function shows Mn $B_g$-derived almost flat bands close to $E_F$, an important question arises; to what extent the nature of our flat $B_g$-bands originates from the kagome lattice physics, namely the frustration-induced destructive interference and the resulting suppression of kinetic energy scale? To answer this, we constructed a set of two electronic Wannier orbitals of $B_g$-bands from our DFT band structure, where each of two Mn kagome layers in the unit cell hosts one $B_g$-orbital. Figure 3i, j show our Wannier-projected bands and the real-space Wannier orbital, respectively (see Fig. 1c for a schematic illustration). As discussed in a previous study on CoSn[17], such an orbital shape with alternating sign at neighboring sites of a hexagon suppresses electron hopping between neighboring sites via destructive interferences induced by geometric frustration, which makes the $B_g$-bands narrower and susceptible to on-site Coulomb correlations.

The role of electron correlations shown in Fig. 3 (also schematically in Fig. 1e), which selectively promotes kagome bands, is reminiscent of the formation of coherent peak and lower/upper Hubbard bands in the orbital-selective Mott transitions observed in several multiorbital systems such as $Ca_{2-x}Sr_xRuO_4$ and Fe-based superconductors[38–41]. Hence, the flat bands in SMAS arise from an unusual cooperation between kagome-induced kinetic energy quenching and orbital-selective electron correlations.

Lastly, we mention that our DMFT results remain paramagnetic down to $T$ = 116 K when U $\leq$ 8 eV and $J_H$ = 0.8 eV, contrary to our DFT+$U$ results. This is somewhat consistent with the experimental observation of no long-range order. To check the ferromagnetic instability at the low-temperature regime, which is beyond the power of the quantum Monte Carlo impurity solver, we employed a rotationally-invariant slave boson methodology combined with DFT (DFT+RISB). DFT+RISB method has been known to capture the correlation-induced band renormalization of the so-called coherent peak close to the Mott

transition, and has been used to study electronic structures of various correlated metals at the zero-temperature limit[64–66]. We checked that DFT+RISB reproduces the essential feature of DMFT results, namely the energy renormalization and band flattening of the $B_g$-bands. A remarkable observation is that, while DFT+RISB results remain para-magnetic most of our choices of $U$ and $J_H$, ferromagnetism emerges only when the $B_g$-bands gets very close to the Fermi level (please refer to Supplementary Note 9 for further details). This is an evidence that the presence of the flat $B_g$-bands in the vicinity of the Fermi level is the origin of the observed ferromagnetic fluctuations in the low-temperature regime.

## Dependence of $B_g$ energy renormalization on $U$, $J_H$, and double-counting energy

The earlier analysis of DMFT results at $T$ = 210 K and with $(U, J_H)$ = (6, 0.8) eV revealed that the presence of electron correlation shifts the flat band closer to the $E_F$. In order to understand such nontrivial behavior, we investigated the dependence of band evolution on the change of the on-site Coulomb repulsion $U$, impurity temperature $T$, double-counting energy of Mn $d$-bands, and Hund's coupling $J_H$ within our DMFT method. In this section, we focus on the effects of $U$, $T$, and double-counting shift, while the role of $J_H$ will be discussed in the next section.

Figure 4 summarizes the results by plotting spectral functions and PDOS for three different values of U = 4, 6, 8 eV with a nominal occupancy $n$ = 5.0 in the nominal double-counting scheme[67] employed at $T$ = 500 and 1500 K. Here, the nominal occupancy describes the position of the correlated Mn $d$ band in energy, which can be shifted via tuning the value of $n$ (see Supplementary Note 7 for results from other choices of $n$). As shown in Fig. 4a-c, enhancement of $U$-value from 4 to 8 eV makes the flat $B_g$ bands move towards $E_F$ with the bandwidth more suppressed, while the positions of $A_g^{in}$ and $A_g^{out}$ peaks in the PDOS plots are left almost unaffected. Figure 4g, h depict the mass enhancement and on-site energy renormalization (i.e., $\mathrm{Re}\Sigma(\omega=0)$) induced by $U$ at $T$ = 500 K, respectively, while Fig. 4i, j show the same quantities as a function of temperature at $U$ = 6 eV. These data show that both $U$ and temperature affect the energy renormalization of the $B_g$-orbitals, and that orbital-dependent mass enhancement of the $B_g$ and $A_g^{in}$ are stronger than that of $A_g^{out}$.

In Fig. 4c we observe broad humps of $B_g$ states located around $\pm 0.5$ eV with respect to $E_F$, which can be identified as the lower and upper Hubbard bands originating from the on-site $U$. The energy

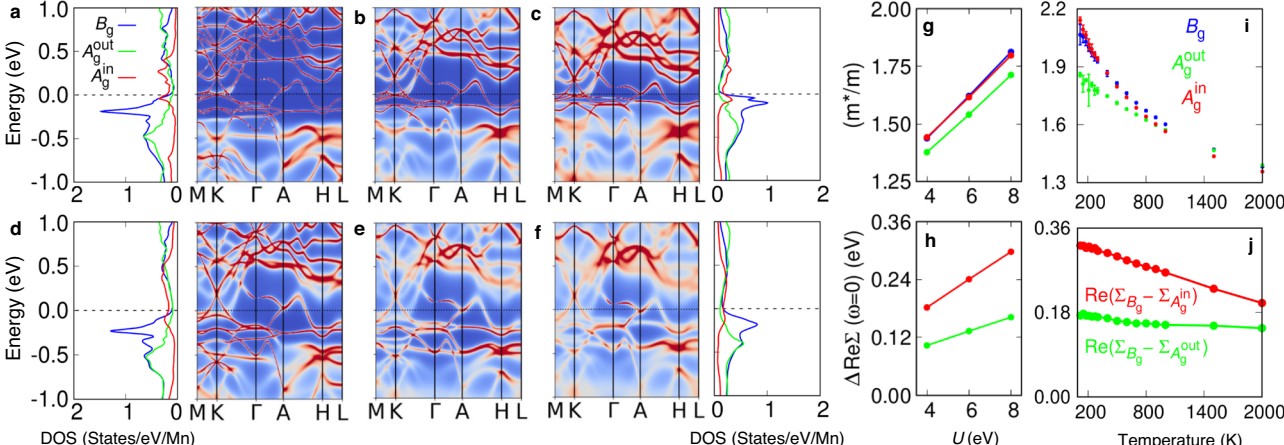

**Fig. 4 | $U$- and $T$-dependence of orbital-dependent band renormalization.**
**a–c** and **d–f** Spectral functions plotted at 500 and 1500 K with three sets of $(U, J_H)$ = (4, 0.8), (6, 0.8), and (8, 0.8) eV, respectively. The PDOS are plotted only for four different cases and shown on the left and right side of the spectral functions. **g** The mass enhancement as a function of on-site Coulomb repulsion $U$. **h** On-site energy renormalization: $\mathrm{Re}(\Sigma_{B_g} - \Sigma_{A_g^{out}})$ (green) and $\mathrm{Re}(\Sigma_{B_g} - \Sigma_{A_g^{in}})$ (red) versus $U$ at zero

frequency. **g, h** The results are presented at $T$ = 500 K. **i** Mass enhancement plotted against temperature with error bars for $(U, J_H)$ = (8, 0.8) eV. **j** On-site energy renormalization: $\mathrm{Re}(\Sigma_{B_g} - \Sigma_{A_g^{out}})$ (green) and $\mathrm{Re}(\Sigma_{B_g} - \Sigma_{A_g^{in}})$ (red) versus temperature at zero frequency. All calculations shown in the panels above were obtained using full Coulomb interaction.

difference between the upper and lower Hubbard bands (~1 eV) is a fraction of the magnitude of the on-site Coulomb $U$ (4 ~ 8 eV in Fig. 4) parameter, which can be attributed to the formation of $B_g$ molecular orbitals and the resulting renormalization of $U$ within the molecular orbital sector[68]. Together with the $U$-induced emergence of a sharp coherence peak at the $E_F$[62,63], this phase can be considered an incipient orbital-selective Mott phase. Note that similar orbital-selective incipient Mott phase in flat-band systems has been reported in a kagome-induced lattice model[59,60]. However the entrance to true orbital-selective Mott-insulating phase is arrested by the presence of other weakly-correlated bands, strong Mn-Si hybridization, and non-negligible inter-orbital hopping channels[44,45,69].

Secondly, in order to see the effect of temperature, in Fig. 4d–f we plot spectral functions at $T = 1500$ K ($U = 4$, 6, and 8 eV, respectively). Comparison with the $T = 500$ K results (Fig. 4a–c) reveal that an increase in temperature tends to cancel the $U$-induced renormalization of $B_g$-bands. The $T$-induced shifting down of $B_g$-bands is most significant at $U = 8$ eV, where the renormalization of the $B_g$-bands is the strongest, and almost negligible in less-correlated cases with $U = 4$ eV, aside from a trivial $T$-induced overall blurring of spectra. This $T$-induced evolution of electron correlations can also be checked in the $T$-dependent mass enhancement as shown in Fig. 4i, where the orbital differentiation between $B_g$, $A_g^{in}$, and $A_g^{out}$ orbitals become significant below $T = 500$ K. Note that the mass enhancement of $A_g^{in}$ orbital is also comparable to that of $B_g$, but without a proper kagome-induced destructive interference, it does not exhibit any significant correlation-induced changes in the spectral functions.

Finally, we comment that the change of $U$ and $T$ shows a common feature with respect to the renormalization of the $B_g$ bands. As $U$ is enhanced or $T$ is lowered, the Mn $d$-orbital occupation reduces and gets closer to 5. For example, at the nominal charge of $n = 5.0$ and at $T = 500$ K, increasing the value of $U$ from 4 to 8 eV reduces the $d$-occupancy from 5.48 to 5.33. On the other hand, at $U = 8$ eV, cooling the system from $T = 1500$ to 500 K, reduces the $d$-occupancy from 5.34 to 5.32. Although the change in the Mn $d$ occupancy in solid is not dramatic, it follows the same trend. This observation is consistent with a previous theoretical result, where the orbital-selective correlation effect is found to become stronger as the system approaches the half-filling regime[40,41]. Shifting the entire Mn $d$-orbitals in energy by tuning the nominal charge $n$ further confirms this observation; pushing the $d$-orbitals downward makes their $d$-orbital occupancy enhanced and tends to remove the orbital-dependent correlation effects, and vice versa (See Supplementary Note 7 for further details).

## Orbital decoupling and bad metallic phase by Hund's coupling

So far, the DMFT results with a fixed value of Hund's coupling $J_H = 0.8$ eV have been presented. In $d$-orbital systems like SMAS understanding the role of Hund's coupling is essential because (i) orbital-selective Mott character, which is essential to the emergence of the $B_g$ kagome bands, has been reported to be strongly enhanced by the Hund's coupling[44,45], and (ii) in the $d^5$-limit, where the effects of Hund's coupling is the strongest[49], the most strongest orbital-selective correlation effects are observed both in our results and previous studies[40,41].

We begin with presenting the probability distribution of impurity multiplet states from the Monte Carlo solver with $(U, J_H) = (8, 0.8)$ eV at $T = 300$ K as depicted in Fig. 5a. The predominance of high-spin configurations in each charge occupation sector is clear, which overall yields the estimate of the size of the Mn moment to be 0.90 $\mu_B$/Mn. Increasing $J_H$ up to 1.0 eV induces strong blurring of band dispersions, which can be attributed to the enhancement of local moment-induced scattering, where the size of the moment also increases up to 1.16 $\mu_B$/Mn at $J_H = 1.0$ eV. Because the Mn moment 0.86 $\mu_B$/Mn is deduced from our Curie-Weiss fit, we concluded that $J_H = 0.8$ eV is a reasonable choice and adopted this value for obtaining most of the presented results in this work unless specified.

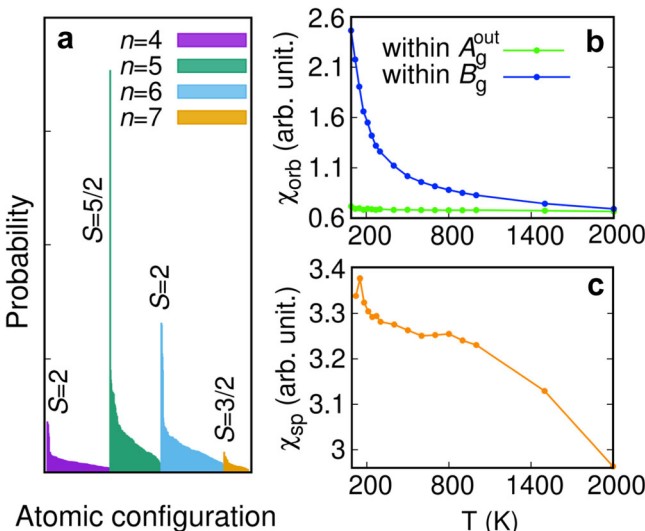

**Fig. 5 | Tendency toward local moment formation and DC susceptibilities.** **a** Histogram of Mn $d$ atomic configurations showing the probability in descending order. For each $n$, high spin configuration carrying maximum probability is marked. Ising-type Coulomb interactions were employed. **b**, **c** DC orbital and spin susceptibilities versus temperature plot, computed for $(U, J_H) = (8, 0.8)$ eV using full Coulomb interaction. The figure legends, within $B_g$ and $A_g^{out}$ stand for difference in orbital susceptibilities within $B_g$ and $A_g^{out}$ orbital sectors, respectively.

The nature of the Hund's-coupling-induced incoherent metallic phase can be further substantiated by the imaginary part of Mn self-energy in the Matsubara frequency axis. By examining the power-law behavior of imaginary part of Mn self-energy $-\text{Im}\Sigma(\beta\omega_n) \sim \gamma + K\omega_n^\alpha$ at $(U, J_H) = (8, 0.8)$ eV, we investigate the $J_H$-induced deviation of our system from Fermi-liquid-like behavior. Here, $\gamma$ ($\equiv -\text{Im}\Sigma(\omega_n \to 0)$) stands for the low-frequency scattering rate and $\omega_n$ ($= (2n + 1)\pi T$) is the Matsubara frequency with $\alpha$ being exponent. From the fitting, the exponent $\alpha$ is found to vary between 0.35–0.39 for a wide temperature range $T = 120$–2000 K, implying a significant deviation from Fermi-liquid behavior ($\alpha = 1$)[70]. From low-frequency DMFT self-energy, we computed band renormalization factor ($Z^{-1}$) and mass enhancement ($m^*$) for each orbital. The mass enhancement and renormalization factor are connected by the following equation, $m^*/m = Z^{-1} = 1 - \partial\text{Im}\Sigma(i\omega)/\partial\omega|_{\omega \to 0^+}$[71]. In the studied $T = 120$–2000 K range, we observe three different $m^*/m$, viz., 2.07–1.38 for $B_g$, 1.85–1.39 for $A_g^{out}$, and 2.14–1.35 for $A_g^{in}$ orbitals (see Fig. 4i). From this result, we conclude that $B_g$ and $A_g^{in}$ show clear orbital-dependent correlations in comparison to $A_g^{out}$, driven by the Hund's coupling[72].

In addition to the incoherent metallicity, another major role of Hund's coupling is quenching the orbital degree of freedom and decoupling orbitals, thereby enabling orbital-dependent correlation effects[44,45]. An additional role is inducing the so-called Hund's metal phase[47], where the spin and orbital degrees of freedom decouple and the orbital degree of freedom is more quenched at higher energy than the spin one[73]. To get insight into these aspects, we computed static spin and orbital susceptibilities as functions of temperature using the following formulae, $\chi_{sp} = \int_0^\beta \langle S_z(\tau)S_z(0)\rangle d\tau$ and $\chi_{orb} = \int_0^\beta \langle \Delta N_{orb}(\tau)\Delta N_{orb}\rangle d\tau - \beta \langle \Delta N_{orb}\rangle^2$, where $S_z$ is the total spin angular momentum of Mn $d$ orbitals and $\Delta N_{orb} = N_a - N_b$ is defined as the occupation difference between two orbitals (or the difference in average occupations of the two groups of orbitals) within the chosen orbital multiplet[74]. In other words, we are interested in charge fluctuations within an orbital sector at our choice, and contrasting behaviors of $\chi_{orb}$ depending on the choice of orbital sectors can be a signature of orbital differentiation between them. In Fig. 5b, the DC orbital susceptibilities ($\omega = 0$) for $A_g^{out}$ and $B_g$ orbitals are

plotted as functions of temperature. The $\chi_{orbs}$ for $A_g^{out}$ and $B_g$ indeed show stark contrast; while $\chi_{orb}$ for $A_g^{out}$ orbitals remains almost constant, that for $B_g$ shows sharp enhancement as $T$ is lowered. This shows that the two orbital sectors are decoupled by $J_H$, and that the correlation-induced localization of electrons, via orbital-selective correlations and the emergence of correlation-induced $B_g$ kagome bands, induces strong charge fluctuations within the $B_g$ sector. Note that in the presence of spin-orbit coupling, this can lead to enhancement of spin fluctuation within the $B_g$ sector as well.

In Fig. 5c, the DC spin susceptibility is plotted as a function of $T$. We observe that with decreasing temperature, the spin susceptibility gradually enhances. Because of computational cost issue, we could not lower the temperature below $T = 116$ K, so the quenching of orbital degree of freedom (*i.e.* peak of orbital susceptibility) prior to the spin one with decreasing $T$ could not be captured in this study. In another study on the three-orbital Hubbard model, it has been argued that Mott physics is more dominant closer to the half-filling limit[73]. For a better understanding of the nature of the low-$T$ phase of SMAS, further studies are necessary in the near future.

### Signatures of flat bands in magnetoresistance and optical conductivity

Signatures of correlation-induced flat dispersions can be further explored through magnetoresistance (MR) signals. Figure 6a, b present the temperature dependence of a transverse MR ratio measured in the field range of $-9 < B < 9$ T for $H//a$ and $H//c$ orientations. Below 11 K, a negative MR is observed with its magnitude rapidly increasing as the temperature is lowered. This negative MR grows without saturation in fields up to 9 T and temperatures down to 2 K. Noteworthy is that the emergence of the negative MR effect is correlated with the observed upturn in resistivity (see Fig. 2a) and the Brillouin scaling, specifically the amplitude parameter $A(T)$ exhibited in Fig. 2f. The similarities between $A(T)$ and MR($T$) suggest that the negative MR signal is linked to the formation of flat bands, which promote ferromagnetic fluctuations. Further evidence is seen in the deviation of MR($B$) from the conventional $B^2$ (or $B^{1.5}$) dependence at fields above 4.5 T, as depicted by the dotted and solid lines in Fig. 6a, b. However, the $B^{-1/3}$ dependence typically expected for nearly ferromagnetic materials could not be identified within the measured field range up to 9 T, possibly due to the moderate ferromagnetic correlations[75]. Additional high-field measurements are essential to definitively corroborate this dependence.

The optical conductivity shown in Fig. 7a (see Supplementary Note 3 for more details) further provides evidence for enhancement of the electron scattering below 30 K. Below 1500 cm$^{-1}$, where the intra- and interband transitions are roughly divided, a spectral weight

transfer from high- to low-frequency regimes occurs as the temperature is lowered, resulting in the sharpening of the Drude peak. But, there is a slight suppression of the Drude peak below 30 K, as can be seen in the inset of Fig. 7a, where the DC resistivity extracted by the extrapolation of the optical conductivity ($\sigma_1(\omega)$) to zero frequency is shown; the DC resistivity exhibits a slight upturn below 30 K, which is consistent with the transport measurement (see Fig. 2a). The optical scattering rate (see Supplementary Note 3) presents a similar trend. Such a trend is observed in our computed optical conductivity shown in Fig. 7b, where the optical conductivities were obtained from DFT and DMFT ($U = 8$ eV and $J_H = 0.8$ eV) results. The DMFT optical conductivity spectra show similar frequency- and temperature-dependent behaviors as the optically measured ones; both optical conductivity spectra show a broad dip near 1500 cm$^{-1}$. Note that the suppression of the Drude peak by the inclusion of dynamical correlations is noticeable by comparing DFT (black curve) and DMFT (colored curves), which is consistent with the large values of the effective mass (see Fig. 4g, i, and Supplementary Note 3). DMFT $\sigma_1(\omega)$ also shows the enhancement of electron scattering as the temperature is lowered from 210 to 120 K; a slight suppression of $\sigma_1(0)$ between $T = 210$ and 120 K is in qualitative agreement with experimental observations (note that the temperature scale is overemphasized in DMFT, where only the electronic temperature contributions at impurity sites are incorporated). The suppression of the Drude contribution in the low temperature regime can be attributed to the enhancement of orbital fluctuations, as shown in Fig. 5b, which may lead to the enhanced magnetic fluctuations as the spin-orbit coupling is included.

## Discussion

From our experimental results, we observed several interesting phenomena at low-temperature regimes, such as an upturn in resistivity below 30 K, deviation from Curie-Weiss behavior below 100 K, the power-law behavior of the magnetic susceptibility and internal fields suggesting ferromagnetic fluctuation, and non-saturating negative magnetoresistance down to $T = 2$ K. On the other hand, studying low-temperature phenomena below 100 K is computationally limited to our case due to the computational costs of CTQMC solver in the presence of significant hybridizations.

Nevertheless, our DMFT calculations reveal an unexpected emergence of kagome-induced flat band physics via electron correlations, which seems the only viable way to understand the observed ferromagnetic fluctuations. We speculate that as the temperature is lowered below 100 K, the flat band may shift even closer to $E_F$ and may lead to various electronic instabilities including ferromagnetic ones as suggested in another kagome magnet FeSn[76,77]. Indeed, our DFT

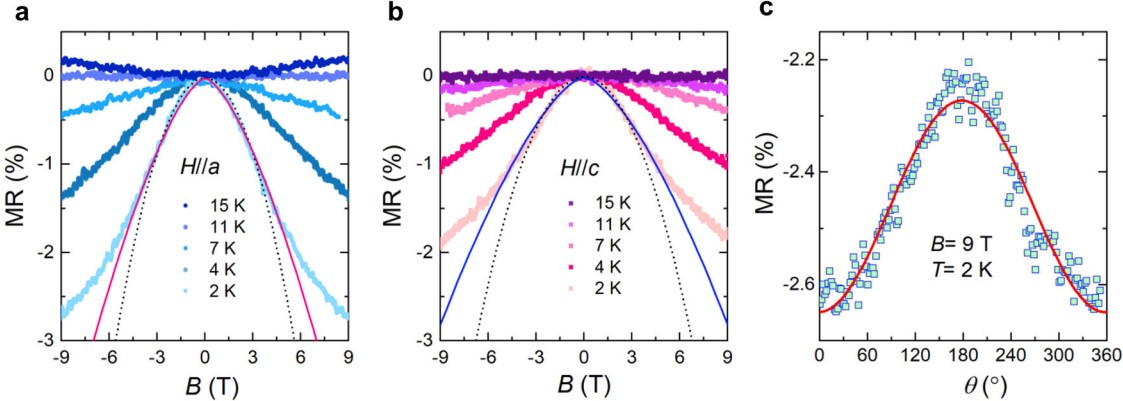

**Fig. 6 | Negative magnetoresistance signature.** Transverse magnetoresistance of SMAS measured at selected temperatures with a magnetic field applied to $H//a$ **a** and $H//c$ **b**. The dotted lines represent fits to a $B^2$ dependence and the solid lines to a $B^{1.5}$ dependence of the low-field magnetoresistance. **c** Angle-dependent magnetoresistance measured at $T = 2$ K in a magnetic field of $B = 9$ T.

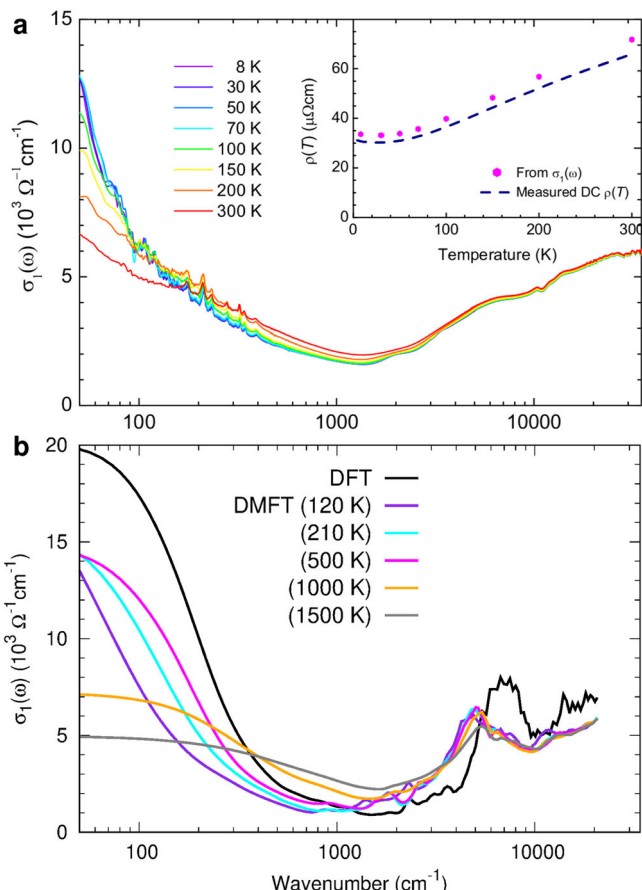

**Fig. 7 | Comparison between experimental and computed optical conductivities. a** Experimental optical conductivity of SMAS at various temperatures. In the inset, the DC resistivity obtained from the optical conductivity and measured DC resistivity are compared. **b** Computed optical conductivity from DMFT calculations for $(U, J_H)$ = (8, 0.8) eV. In both cases normal incidence is considered.

+RISB[64,65] result shows that the presence of the flat bands close to the Fermi level can create the ferromagnetic instability, supporting our speculation above. Further investigation is necessary to explore the nature of the low-$T$ ground state of this system.

There are several heavy-fermion compounds such as CeRh$_6$Ge$_4$, which have both the flat and dispersive bands at the Fermi level that host ferromagnetism[78–80]. Despite differences in chemical compositions and correlated subspaces involved ($d$ vs. $f$), there has been a growing interest in the universality between $f$-orbital-based Kondo systems and $d$- or $p$-orbital-based kagome flat band systems, where the heavy electron bands in f-based Kondo systems are analogous to the flat bands in d-based kagome lattices or even $p$-orbital-based twisted bilayers[81,82]. Given that our system shows a large Sommerfeld coefficient[52], and power-law behavior of magnetic susceptibility and specific heat in the low-temperature regime (below 10 K), we believe that our Mn-kagome Sc$_3$Mn$_3$Al$_7$Si$_5$ may share an interesting universality with a broader class of correlated materials.

Finally, we comment that a recent theoretical study suggests that the orbital-selectivity of electron correlations found in our system can be a general phenomenon in transition-metal-based kagome metal systems, where nearly flat kagome bands coexist with wide dispersive bands (such as ligand-originated bands or ones irrelevant to kagome-induced kinetic energy quenching)[60,83]. Additionally, it has been also suggested that many kagome metals may host universal long-range Coulomb interactions. In combination with the SOC-induced gap opening and the wider spread of Berry curvature over momentum

space induced via flat dispersion, one may ask about the possibility of realizing interesting correlated phenomena such as fractional Chern insulators[3,5,21] and Weyl-Kondo semimetal phase[84] on top of the on-site-correlations-induced flat bands in SMAS.

In summary, we have investigated the nature of electronic correlations and magnetic properties of Mn-based kagome metal Sc$_3$Mn$_3$Al$_7$Si$_5$, combining a multitude of experimental and theoretical techniques. The temperature and field-dependent magnetization measurement signifies the presence of ferromagnetic fluctuations at very low temperatures. The upturn in the resistivity alludes to the development of electron correlations below 30 K. The dynamical mean-field calculations reveal correlation-induced flat bands close to $E_F$ at $k_z = 0$ with an additional nodal surface band at $k_z = \pi$ guaranteed by nonsymmorphic twofold screw rotation and time-reversal symmetries. With the inclusion of spin-orbit coupling, a gap opens up at the Dirac points and the flat bands are likely to become topologically nontrivial. Therefore, SMAS constitutes a potentially promising platform to explore the interplay between electron correlations and SOC in kagome flat band systems.

## Methods

### Sample synthesis, magnetic and transport properties

High-quality SMAS single crystals were prepared using the self-flux method. The magnetic measurements were performed using a superconducting quantum interference device vibrating sample magnetometer (SQUID VSM). The NMR measurements were done by employing a home-made NMR spectrometer and an Oxford Teslatron PT superconducting magnet. The magnetoresistance (MR = [$\rho(B) - \rho(0)$]/$\rho(0)$) measurements were performed at ambient pressure using the electrical transport option of the Quantum Design Physical Properties Measurement System with a four-point contact configuration. To measure reflectance at various temperatures, a commercially available spectrometer Vertex 80v and a continuous liquid helium flow cryostat were used.

### Electronic structure calculations

For an accurate and appropriate treatment of dynamic electron correlations in the electronic structure of SMAS, an ab-initio density functional theory (DFT) and dynamical mean-field theory (DMFT) methods were employed. The DFT calculations were carried out within the framework of local density approximation (LDA)[85], using a full potential linearized augmented plane wave plus local orbital (LAPW+lo) method. For investigation of dynamical correlation effect, a fully charge-self-consistent DMFT method, as implemented in the embedded DMFT functional code[67,86], was employed in combination with the WIEN2K package[87]. Throughout the entire manuscript, rotationally-invariant full Coulomb interactions were used for the impurity problem unless otherwise specified. In some cases Ising-type density-density interactions were adopted. We checked that the choice of the Coulomb interactions does not affect our core results. A nominal double-counting scheme[67] with the Mn nominal charge $n$ = 5.0 was adopted, where the validity of the double-counting parameter was justified by comparison to results from an exact double-counting scheme[88] (see Supplementary Note 7 and Supplementary Fig. 9 therein for more details). For our DFT +RISB calculations, we employed CYGUTZ (https://cygutz.readthedocs.io/) package in combination with WIEN2K[64,65]. $RK_{max}$ = 9.0 was employed, and for a better convergence, a non-shifted $k$-grid of up to $17 \times 17 \times 14$ was used. Mn $d$-orbital was set to be the correlated active subspace. More details on our experimental methods and computational parameters are listed in Supplementary Note 1 and 2, respectively.

## Data availability

The data supporting the findings of this study are available within the manuscript and Supplementary Information. Additional relevant data are available from the corresponding authors upon request.

## Code availability

The Vienna ab initio Simulation Package (VASP, Ver. 6.3, see https://www.vasp.at) and WIEN2K (Ver. 2019, see http://www.wien2k.at) are commercial codes, while DFT+embedded DMFT Functional code (see http://hauleweb.rutgers.edu/tutorials/) is an open-source one which runs on top of the WIEN2K package.

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

## Acknowledgements

We thank Bohm-Jung Yang, Chang-Jong Kang, Sangkook Choi, Ara Go, Choong Hyun Kim, Minjae Kim, Seung-Sup B. Lee, and Kristjan Haule for fruitful discussions. This work was supported by the Korea Research Fellow (KRF) Program and the Basic Science Research Program through the National Research Foundation of Korea funded by the Ministry of Education [Grant No. NRF-2019H1D3A1A01102984, NRF-2020R1C1C1005900, RS-2023-00220471], and also the support of computational resources including technical assistance from the National Supercomputing Center of Korea [Grant No. KSC-2021-CRE-0222, KSC-2022-CRE-0358]. HSK additionally appreciates the Asia Pacific Center for Theoretical Physics (APCTP) and the Center for Theoretical Physics of Complex Systems at the Institute of Basic Science (PCS-IBS) for their hospitality during the completion of this work. The work at SKKU is supported by the National Research Foundation (NRF) of Korea [Grants No. 2020R1A2C3012367, 2020R1A5A1016518, 2021R1A2C101109811, and 2022H1D3A3A01077468]. Work by YY was supported by the U.S. Department of Energy (DOE), Office of Science, Basic Energy Sciences, Materials Science and Engineering Division, including the grant of computer time at the National Energy Research Scientific Computing Center (NERSC) in Berkeley, California. This part of research was performed at the Ames National Laboratory, which is operated for the U.S. DOE by Iowa State University under Contract No. DE-AC02-07CH11358.

## Author contributions

J.H., K.-Y.C. and H.-S.K. designed and supervised the overall research. S.S. performed electronic structure calculations. P.P., C.L., S.J. and H.C. conducted sample preparations and experimental measurements. Y. Y. provided support in installing CYGUTZ package and performing DFT+RISB calculations. All authors discussed the results and contributed to writing the manuscript.

## Competing interests

The authors declare no competing interests.
