## [Peer Review File · Nature Communications]

REVIEWER COMMENTS

Reviewer #1 (Remarks to the Author):

Systems having electronic flat-bands are promising candidates to explore strongly correlated phenomena in condensed matter physics. Several multiorbital systems have been reported recently, having large momentum distribution of such bands with small bandwidths. While in 2D single-orbital flat band systems the ground state has been shown to exhibit ferromagnetism, signatures of correlation driven states have been rarely observed in multiorbital systems with co-existing flat and dispersive bands. In this regard, the present work by the authors is noteworthy, where they have claimed that the system $\text{Sc}_3\text{Mn}_3\text{Al}_7\text{Si}_5$ is at the verge of ferromagnetic instability originating from correlation driven selective multiorbital flat bands.

The results presented in the manuscript would encourage investigations into tuning of the Fermi level energy with respect to the flat bands (via chemical doping/strain), and would thus be beneficial towards further understanding of electronic correlation driven effects in multiorbital flat-band systems. Similar attempts have been made for flat-band system CoSn recently [J. Zhang et al. arXiv:2105.08888 (2021)] as cited in the manuscript, and further investigations into $\text{Sc}_3\text{Mn}_3\text{Al}_7\text{Si}_5$ in future would be useful.

Magnetic susceptibility, NMR and magnetoresistance measurements presented in the present article show ferromagnetic signatures below 25 K in $\text{Sc}_3\text{Mn}_3\text{Al}_7\text{Si}_5$. This is a surprising result given the frustrated Kagome geometry and large negative Curie Weiss temperature of 262 K and may hint towards the dominance of flat-band driven effects at low temperatures over the frustration induced antiferromagnetic-like fluctuations. Could the authors comment further on this competition?

The authors have shown that LDA+U calculations are unable to account for the ferromagnetic instabilities. Instead, LDA+DMFT calculations show the formation of dynamic correlation driven flat-bands for selective orbitals close to the Fermi level. They have demonstrated the origin of the flat-bands to be two-fold: Kagome and correlation driven. These are interesting observations which emphasise the importance of considering multiorbital nature of correlated orbitals with orbital selectivity in understanding the topological band properties of such materials, as also demonstrated by H.Hu et al. arXiv:2209.10396 (2022), Z. Liu et al. Nat. Commun. 11, 4002 (2020) and D. Di Sante et al. Phys. Rev. Res. 5, L012008 (2023), and cited by the authors.

While there is no direct experimental evidence of band flattening in the present work, the theoretical results would however, simulate future experimental investigations of momentum space flat-bands in such systems and may provide concrete evidence if the correlation driven flat-band physics is solely and directly responsible for the ferromagnetic tendencies at low temperatures.

However, there are a few critical questions towards the key message which prevent us for the recommendation:

1. The authors claim that, the ferromagnetic instabilities arise from the band flattening and their increasing proximity to the Fermi level as the temperature is lowered in the system. Theoretically, they show selective band flattening, shift towards Fermi level, and enhancement of spin susceptibility. Still, a conclusive evidence of flat-band induced experimentally observed magnetism is lacking. For example, could the authors show enhancement in the degree of spin polarisation and account for the amount of magnetoresistance observed?

2. If as the authors claim, the Bg orbitals are selectively responsible for the enhancement in the ferromagnetism, do they observe direction dependent change in magneto resistance. In the manuscript they have presented magnetoresistance with magnetic field in the Kagome plane (no specific direction has been measured). Does the magnetoresistance have angular dependence in the ab plane or to the c axis? Are the ferromagnetic fluctuations in the Kagome plane or out of plane? Could the authors also please mention the direction of applied magnetic field in the magnetic susceptibility and MH measurements? In addition, do they observe anisotropic behaviour in the magnetism at low temperatures in the in-plane and out-of-plane directions?

3. Describing Fig. 4g the authors state that that orbital-dependent mass enhancement and shift of $\text{Re}\Sigma(0)$ are strongest in the Bg orbitals. This is clearly not the case in Fig. 4g where Bg and also the authors note later in the manuscript that the mass enhancement of Agin orbital is also comparable to that of Bg. These statements need further clarification and consistency. Also, the authors do not show independently how the energy renormalisation behaves for Agin orbitals w.r.t. U, and a comparison of energy renormalisation vs temperature between Bg and Agin orbital. A substantial difference may help elucidate their claim of dominant role of Bg orbital, despite its comparable mass enhancement as Agin orbital. Also, the authors do not show the orbital susceptibility change as a function of temperature for the Agin orbital in Fig. 5b.

4. It has been concluded in the manuscript that the resistivity upturn at low temperatures is a result of enhanced electronic scattering, associated with development of ferromagnetism in the system. With the development of ferromagnetism, one would expect increase in spin polarisation and hence smaller electron-electron scattering and lower resistivity. Could the authors explain this observation?

Further comments:

It might be helpful for readers, if the authors can include in Figure 1, schematically, the five local d orbital orientations w.r.t. the Kagome planes in addition to the coloured arrows used to represent the axes at present.

The authors mention, 'Below 1500 cm^{-1} , where the intra- and interband transitions are roughly divided, a spectral weight transfer from low- to high-frequency regimes occurs as the temperature is lowered, resulting in the sharpening of the Drude peak.' It appears however, as if spectral weight transfer is transferred from high- to low-frequency regimes as the temperature is lowered, which would indicate the gradual band movement towards the Fermi level.

Reviewer #2 (Remarks to the Author):

Reviewer #3 (Remarks to the Author):

In this work, the authors employed a collection of experimental and theoretical techniques to characterize the electronic properties of $\text{Sc}_3\text{Mn}_3\text{Al}_7\text{Si}_5$ (SMAS), which contains a kagome substructure of Mn atoms. Experimentally, the authors observed the presence of ferromagnetic fluctuations at low temperatures in spite of a negative Curie-Weiss temperature. This was inferred from a small magnetization hysteresis and ^{27}Al NMR. The effective magnetic moment found was also substantially smaller than the spin-only value expected for a Mn $3+$ ion, which indicated itinerant magnetism. Additionally, the authors observed a significant negative magnetoresistance and a slight suppression of the Drude peak in the optical conductivity, which were interpreted as being consistent an enhancement in electron scattering due to the ferromagnetic fluctuations. On the theory front, the authors first identified a set of relatively flat bands from the DFT band structure and confirmed their Wannier functions resemble the shape expected for the flat bands arising from a kagome structure. However, these flat bands are energetically quite far away from the Fermi level. The authors then showed that electronic correlations could bring these flat bands closer to the Fermi level as the system becomes a correlated metal which is incipient to an orbital selective Mott phase, in the sense that incoherent spectral weight associated with the upper and lower Hubbard bands have developed but a coherent spectral peak remained. Further evidence was provided to support the orbital dependence on the correlation effect.

Overall, this manuscript presented a collection of interesting experimental phenomena observed at low temperature in SMAS and combined that with a theoretical study on the expected effects of electronic correlation on this compound. While these results are certainly interesting in their own rights, this referee judged that the two aspects of the study, namely experimental and theoretical results, were not truly tied together. More specifically, the experimental results all correspond to (indirect) evidence for the presence of significant ferromagnetic fluctuations at low temperatures, whereas the theoretical results indicated a flattening and shifting of the Bg-bands as a set of flat bands at the Fermi level. The relation between the theory and the experiments then rest upon the idea of flat-band ferromagnetism, i.e., the assumption that electrons occupying a flat band would tend to develop ferromagnetic order. While this is a believable explanation of the experimental results, it is also a rather indirect conclusion and there are many subtleties one should be concerned with in pursuing this interpretation. For instance, as the authors noted the kagome flat bands arise

from geometrical frustration. It is unclear to the present referee if one could immediately generalize the existing results on flat-band ferromagnetism to such a setup involving nontrivial wave functions. The presence of a large number of additional bands also raise question on whether or not one can immediately expect a simple relation between the correlated flat bands and ferromagnetic fluctuation. Also, due to computational cost the DMFT studies were restricted to temperatures which were one order of magnitude higher than that relevant for the experimentally interesting observations. Understandably, one could only draw general conclusions on the trend of, for instance, the development of orbital-selective correlation etc. In the opinion of the referee, however, it would be risky to take these general trends as support for an experimental proof of observing correlation-induced flat bands near the Fermi energy.

All in all, while this manuscript certainly highlight SMAS as a potentially interesting material which warrants further investigations, to meet the high standards of Nature Communications, the referee believes a much stronger link between the theoretical and experimental studies presented around SMAS would need to be established.

Reviewer #4 (Remarks to the Author):

In this manuscript the authors explore the electronic properties of the Kagome materials $\text{Sc}_3\text{Mn}_3\text{Si}_5\text{Al}_7$ through a combination of susceptibility, NMR, transport and optics measurements as well as band calculations.

There has been a lot of recent interest in metallic Kagome systems. Unlike the previous focus on the magnetic frustration of already localized spins such as what is found in the insulating Kagome materials Herbertsmithite, the interest in metallic Kagome systems has focused on the electronic structure and the formation of flat bands due to the Kagome structure. The flat bands in Kagome systems stem from localization of charge due to destructive interference caused by the geometry of the Kagome lattice. This localization and formation of flat bands is an important precursor for several interesting topological and non-topological phases. Therefore the study of new flat band metallic Kagome materials is a relevant and important endeavor.

The authors have a comprehensive study of DFT and DMFT calculations on the formation of flat bands and the orbital contributions that form these flat bands.

The authors show some magnetic susceptibility data which is similar to what has been previously presented and is only tenuous evidence at best for ferromagnetic order.

The authors claim there is a ferromagnetic instability below 30 K, however their Curie-Weiss fit shows a large antiferromagnetic coupling. The authors should comment on the relative energy scale of the ferromagnetic instability at low temperature compared to the mean-field antiferromagnetic coupling at high temperature.

The authors present as well measurements of the magnetoresistance and optical conductivity. For those measurements they comment it is consistent with their calculations but provide little more conclusions to draw from it.

Specific comments on the paper

In page 2 second to last paragraph, the authors claim that the magnetic susceptibility is a sign of weak ferromagnetism. 'Weak ferromagnetism' is a feature of the dzyaloshinskii-moriya interaction resulting in a net moment within an otherwise antiferromagnetic material. If the material does not have long range antiferromagnetic order, this would not explain a magnetic hysteresis through weak ferromagnetism. In fact this hysteresis plot is not very convincing, is it sample independent? What is its temperature dependence?

Fig. 2a, and 2b are the same as has been previously reported with similar results.

Fig. 2b, what is the χ_0 used in the Curie Weiss fit and how was it found? This information should be in the Supplementary materials. How does this χ_0 compare in a Sommerfeld-Wilson ratio?

Fig. 2c. Is there really much evidence for a second power law scaling in the range from 10-20K. I point out that any smooth curve will appear linear over a small enough range, which is what appears to be happening in this case.

Fig. 2e, Could this hysteresis be from any Mn impurity on the sample. What is its temperature and sample dependence.

Fig. 3, it would be useful if the Authors included a picture of the Brillouin zone showing the M,K,\Gamma,A,H and L points.

Fig. 3 a-c and e-f show the relative contribution of three different Mn-orbital irreps. However this does not show whether the flat band is predominantly formed from contributions on the Kagome lattice. The authors should include a similar figure (perhaps in the Supplementary materials) showing the relative contribution of each atom to the bands.

Fig. 5c what is the peak at approximately 200 K in the spin susceptibility? Is this a calculation artifact or a real effect?

On the section ORBITAL DECOUPLING AND BAD METALLIC PHASE BY HUND'S COUPLING, the authors say that with a Hund's coupling of 0.8 eV, there is a moment size of 0.9 μ_B at 300 K. Does the increasing orbital moment at lower temperatures imply a larger moment with a larger orbital contribution at lower temperatures. What is the expected moment at 200 K? If the moment is changing between 300 and 200 K, does that fit for the Curie-Weiss fit?

Fig. 6 I point out that the increase in the conductivity at 2 K (Fig. 2a) is approximately 4%, the same quantity that is found in the magnetoresistance. This means that at 9 T, the low temperature upturn in resistivity is likely suppressed.

In the discussion, the authors state that the hysteresis at 2 K is a sign of ferromagnetic fluctuation. I disagree, only an ordered state would have non-zero magnetization at zero field and show that kind of curve. Ferromagnetic fluctuations at zero field will average out to a zero net moment at zero field. Another way to think about this is that in order to have a net ferromagnetic moment at zero field the material needs to have passed a point where the time correlation has diverged, this will only happen after a phase transition has passed.

Comments on the supplementary materials

Section IV the authors mention Korringa law for temperature dependence. They should add a citation explaining what this is.

Section IV. the authors include a number 44.8833 ± 0.2482 . No number should have more than 1-2 digits of uncertainty presented.

Summary of changes made in the revised manuscript

List of changes made are as follows, where new texts inserted are colored in **Red** in the revised manuscript.

1. In the revised Fig. 1, a new panel (c) depicting schematic diagrams of five Mn d orbitals and their relative orientations with respect to Kagome plane is inserted.
2. In Fig. 2 and texts referring to the figure of the revised manuscript, we inserted new data and discussions on the anisotropy of the magnetic signals and developments of multiple energy scales as temperature is lowered.
3. We added field-direction-dependency of our magnetoresistance data in the revised Fig. 6, along with discussions on the field-dependency.
4. In the revised SI (Sec. IX), new slave-boson mean-field calculations results showing the flat-band-induced ferromagnetic instability are included. Relevant discussions are also inserted as the last paragraph in the **Flat Band Realized via Correlation-induced Orbital-dependent Band Energy Renormalization** section.
5. A paragraph suggesting a similarity between our system and ferromagnetic Kondo systems is inserted as the third paragraph in the revised **Discussion** section.
6. We added a new panel (j) in Fig. 4 of the revised manuscript, and also updated relevant texts referring to the figure.
7. Following sentence, “Below 1500 cm^{-1} , where the intra- and interband transitions are roughly divided, a spectral weight transfer from low- to high-frequency regimes occurs as the temperature is lowered, resulting in the sharpening of the Drude peak” in second paragraph under section, *Signature of flat bands in magnetoresistance and optical conductivity*, has been corrected.
8. An inset depicting the hexagonal Brillouin zone and its high-symmetry points is inserted in the revised Fig. 3(a).
9. Discussions on corrected DFT+ U results are appended at the end of the fourth paragraph in the **Flat Band Realized via Correlation-induced Orbital-dependent Band Energy Renormalization** section of the revised manuscript and in the SI (Sec. V).
10. Discussion on new DFT+RISB calculation results are appended as the last paragraph of the **Flat Band Realized via Correlation-induced Orbital-dependent Band En-**

ergy Renormalization section. Also, detailed DFT+RISB results are presented in the SI (Sec. IX).

11. An additional figure, showing individual contribution of each orbital to the flat bands, is added to the supplementary material (Sec. VIII).
12. Spin and orbital susceptibilities shown in Fig. 5(b) and (c) are recomputed for temperature range 120 - 2000 K and updated in the revised manuscript.

Response to REVIEWER COMMENTS

Reviewer #1 (Remarks to the Author):

Systems having electronic flat-bands are promising candidates to explore strongly correlated phenomena in condensed matter physics. Several multiorbital systems have been reported recently, having large momentum distribution of such bands with small bandwidths. While in 2D single-orbital flat band systems the ground state has been shown to exhibit ferromagnetism, signatures of correlation driven states have been rarely observed in multiorbital systems with co-existing flat and dispersive bands. In this regard, the present work by the authors is noteworthy, where they have claimed that the system $Sc_3Mn_3Al_7Si_5$ is at the verge of ferromagnetic instability originating from correlation driven selective multiorbital flat bands.

The results presented in the manuscript would encourage investigations into tuning of the Fermi level energy with respect to the flat bands (via chemical doping/strain), and would thus be beneficial towards further understanding of electronic correlation driven effects in multiorbital flat-band systems. Similar attempts have been made for flat-band system CoSn recently [J. Zhang et al. arXiv:2105.08888 (2021)] as cited in the manuscript, and further investigations into $Sc_3Mn_3Al_7Si_5$ in future would be useful.

Response: We thank Reviewer for appreciating our work and finding it useful for further investigations in future.

Criticism: *Magnetic susceptibility, NMR and magnetoresistance measurements presented in the present article show ferromagnetic signatures below 25 K in $Sc_3Mn_3Al_7Si_5$. This is an surprising result given the frustrated Kagome geometry and large negative Curie Weiss temperature of 262 K and may hint towards the dominance of flat-band driven effects at low temperatures over the frustration induced antiferromagnetic-like fluctuations. Could the authors comment further on this competition?*

Response: We appreciate this insightful comment. As the Reviewer pointed out, a ferromagnetic instability at low temperatures develops out of high-temperature antiferromagnetic interactions deduced from a Curie-Weiss analysis. To obtain a deeper understanding of the thermal evolution of magnetic correlations, we replot our magnetic susceptibilities for H//ab and H//c in a log-log scale in the new Fig. 2(c) in the revised manuscript. Our new plot reveals a multi-stage anisotropic development of magnetic correlations as evidenced by a distinct power-law behavior, for example, $\chi_{ab} \sim T^{-0.17}$ at $T = 25 - 110$ K switches to $\chi_{ab} \sim T^{-0.47}$ below 7 K, while $\chi_c \sim T^{-0.20}$ at $T = 25 - 110$ K to $\chi_c \sim T^{-0.56}$ below 7 K. These thermally varying

magnetic correlations allude to the presence of multiple energy scales, which are associated with the multiorbital character.

In addition, two different but somewhat related perspectives can be addressed in response to the Reviewer's comment. First, some early studies of Hubbard models on frustrated systems show a crossover from the high-temperature paramagnetic Mott insulating phase to the low-temperature metallic phase with a 'coherent peak' feature [Rozenberg *et al.*, *Phys. Rev. Lett.* **75**, 105 (1995), Georges *et al.*, *Rev. Mod. Phys.* **68**, 13 (1996)]. This feature is also observed in our results, for example from the T -dependence presence and absence of the flat bands shown in Fig. 4(b,c,e,f) in the revised manuscript. We argue that, high-temperature Curie-Weiss behavior in our system is overwhelmed by the low-temperature formation of the flat-band Fermi surface and the ferromagnetic (FM) responses induced by the flat bands.

Second, we have found that a similar observation, the suppression of the inverse magnetic susceptibility below a certain temperature, has been observed in a ferromagnetic heavy-fermion compound CeRh_6Ge_4 , which shows a similar crossover from the high- T antiferromagnetic (AF) Curie-Weiss behavior to the suppression of the inverse susceptibility below 30 K, and eventual transition to ferromagnetic phase at $T_C = 2.5$ K [Matsuoka *et al.*, *J. Phys. Soc. Japn.* **84**, 073704 (2015)]. Despite differences in chemical compositions and correlated subspaces involved (d vs. f), there has been a growing interest in the universality between f -orbital-based Kondo systems and d - or p -orbital-based Kagome flat band systems, where the heavy electron bands in f -based Kondo systems are analogous to the flat bands in d -based Kagome lattices or even p -based twisted bilayers [Paschen and Si, *Nat. Rev. Phys.* **3**, 9–26 (2021), Checkelsky *et al.*, arXiv:2312.10659]. Given that our system shows large Sommerfeld coefficient and power-law behavior of magnetic susceptibility and specific heat in the low-temperature regime (below 10 K), we believe that the heavy-fermion-induced ferromagnetic instability in CeRh_6Ge_4 can be relevant to our system.

Changes to the manuscript : In the revised manuscript we updated Fig. 2 to better present the presence of multiple energy scale as shown in the magnetic susceptibility data. Also we inserted a new paragraph on the similarity between our system and ferromagnetic heavy-fermion systems in the **Discussion** section.

The authors have shown that LDA+U calculations are unable to account for the ferromagnetic instabilities. Instead, LDA+DMFT calculations show the formation of dynamic correlation driven flat-bands for selective orbitals close to the Fermi level. They have demonstrated the origin of the flat-bands to be two-fold: Kagome and correlation driven. These are interesting obser-

U (in eV)	$E_{\text{FM}}-E_{\text{AF}}$ (eV/f.u.)	$E_{\text{NM}}-E_{\text{AF}}$ (eV/f.u.)	m_{AF} (μ_{B})	m_{FM} (μ_{B})
1	0.0018	0.0033	0.412	0.133
2	0.1012	0.1216	1.297	0.515
4	0.0143	1.8416	3.102	3.213
6	-0.1382	N/A	3.711	3.742
8	-0.1764	N/A	4.074	4.050

Table 1: Total energies and Mn moment sizes of $\text{Sc}_3\text{Mn}_3\text{Al}_7\text{Si}_5$ from DFT+ U calculations with varying U value.

vations which emphasise the importance of considering multiorbital nature of correlated orbitals with orbital selectivity in understanding the topological band properties of such materials, as also demonstrated by H. Hu et al. arXiv:2209.10396 (2022), Z. Liu et al. Nat. Commun. 11, 4002 (2020) and D. Di Sante et al. Phys. Rev. Res. 5, L012008 (2023), and cited by the authors.

Response: In the previous manuscript we made a mistake in our presentation of DFT+ U calculation results. While the statement by the Reviewer that “LDA+ U calculations are unable to account for the ferromagnetic instabilities” is still valid, our previous claim that LDA+ U calculations do not admit any magnetism is not correct. Please check Fig. 1 and Table 1 for our initial magnetic configurations and the summary of our DFT+ U calculations. In fact, inclusion of the mean-field U value allows both the collinear ferromagnetic and antiferromagnetic solutions, but in the range of $0 < U \leq 4$ eV antiferromagnetic solution is lower in energy than the ferromagnetic solution. Additional calculations by a parameter-free SCAN functional also yield the antiferromagnetic solution to be more stable than the ferromagnetic one. DFT+ U calculations with $U \geq 6$ eV does yield FM solution, but with the size of Mn moments to be $\sim 4 \mu_{\text{B}}$, inconsistent with experimental observations (Please check the above Summary of Changes and the revised SI for more details). Hence simple band calculations and static mean-field treatments of electron correlations cannot account for the observation of the subtle ferromagnetic instability. We apologize all the Reviewers for this confusion, and corrected this issue in the revised manuscript.

Changes to the manuscript : In the **Flat Band Realized via Correlation-induced Orbital-dependent Band Energy Renormalization** section of the revised manuscript, we added a new discussion on our corrected DFT+ U results. Also in the SI we added details on our DFT+ U results.

While there is no direct experimental evidence of band flattening in the present work, the

Figure 1: Schematic illustrations of the ferromagnetic (FM) and antiferromagnetic (AF) initial magnetic configurations we employed in DFT+ U , DMFT, and slave-boson mean-field calculations.

theoretical results would however, simulate future experimental investigations of momentum space flat bands in such systems and may provide concrete evidence if the correlation driven flat-band physics is solely and directly responsible for the ferromagnetic tendencies at low temperatures.

Response: We agree with the Reviewer in totality that the present work does not provide direct experimental evidence of the band flattening phenomenon as observed in the theoretical calculation results. Nevertheless, in the revised manuscript and SI we added a couple of new rotationally-invariant Slave-boson mean-field calculation results showing the tendency toward ferromagnetism at the low-temperature regime. Also, as mentioned above, we comment that our system may share some commonalities with ferromagnetic heavy-fermion systems, supporting the presence of flat-band-induced ferromagnetism. Please check below for further details.

However, there are a few critical questions towards the key message which prevent us for the recommendation:

Criticism-1: *The authors claim that, the ferromagnetic instabilities arise from the band flattening and their increasing proximity to the Fermi level as the temperature is lowered in the system. Theoretically, they show selective band flattening, shift towards Fermi level, and enhancement of spin susceptibility. Still, a conclusive evidence of flat-band induced experimentally observed magnetism is lacking. For example, could the authors show enhancement in the degree of spin polarisation and account for the amount of magnetoresistance observed?*

Response: As the Reviewer and other Reviewers pointed out, the weak link between our DMFT calculation results and our previous manuscript's experimentally observed ferromagnetic instability was the most concerning issue. Difficulties in sample cleaving/polishing for ARPES

or STM quantum interference measurements and approaching the zero-temperature limit in the quantum Monte Carlo calculations within the DMFT calculations prohibited building a direct link between experimental and theoretical results. To tackle this issue, we have followed the steps listed below;

- Confirming the **absence of magnetism in finite-temperature ($T > 116$ K) DMFT calculations**. We checked that, by employing $(U, J_H) = (8, 0.8)$ eV and full Coulomb interactions, DMFT calculations with two different magnetic initial configurations (FM and AFM, see Fig. 1) do not admit any magnetism and converge into the paramagnetic solution above $T = 116$ K. This is in contrast to our DFT+ U calculation results, which stabilize either antiferromagnetic order or ferromagnetism with large Mn moments, but consistent with experimental observations.
- Employment of rotationally-invariant Slave-boson (RISB) mean-field theory in combination with DFT (DFT+RISB) for a **direct check of ferromagnetic instabilities at zero temperature**. DFT+RISB method has been known to capture the correlation-induced band renormalization of the so-called coherent peak close to the Mott transition, and has been used to study electronic structures of various correlated metals at the zero-temperature limit [Lanata *et al.*, *Phys. Rev. X* **5**, 011008 (2015), Lanata *et al.*, *Phys. Rev. B* **87**, 045122 (2013)]. To check whether our system is close to the ferromagnetic instability, we employed CYGUTZ (<https://cygutz.readthedocs.io/en/latest/>) package and made the following observations.
 - We observe the presence of the correlation-promoted flat bands close to the Fermi level with a proper choice of U and J_H , consistently with DMFT results (see Fig. 2 in this letter).
 - The position of the kagome-flat B_g bands is strongly modified by the value of the J_H parameter (also moderately by U), and stronger J_H pushes the flat bands closer to the Fermi level (compare Fig. 2(a) and (b)).
 - The closest agreement with the DMFT bands with $(U, J_H) = (8.0, 0.8)$ eV is achieved when $(U, J_H) = (15.0, 1.5)$ eV was chosen in the DFT+RISB calculations (Fig. 2(b)). Note that this difference between the DMFT and RISB Coulomb parameters is from the limitation of the DFT+RISB methodology, which cannot describe Mott-insulating phase and requires larger Coulomb parameters than realistic ones for systems close to the Mott transition (this has been remedied in the recent ‘ghost-RISB’ method [Lee *et al.*, *Phys. Rev. B* **108**, 245147 (2023)]).

- Nonmagnetic solution is almost always favored over magnetic ones, except when the flat bands are promoted up to the Fermi level by increasing J_H up to 1.8 eV. At $(U, J_H) = (15.0, 1.8)$ eV, a spin splitting of about 0.15 eV occurs at the K-point (see Fig. 2(c)).

Although no signature of ferromagnetic long-range order has been observed in our system, our RISB result shows an example that the flat bands being close to the Fermi level can induce ferromagnetic instabilities, in contrast to DFT+ U results mentioned above, and that the flat bands are indeed located in the vicinity of the Fermi level.

- Searching for **similar examples of ferromagnetic fluctuations in system where flat bands and other dispersive bands coexist** in the vicinity of the Fermi level. Our previous manuscript did not cite much about previous literatures on this point. Therefore after Reviewer’s response we have done literature searches on similar systems with ours and came to realize that, our system shares some characteristics of heavy-fermion systems close to the critical regime, *i.e.* a rather large Sommerfeld coefficient of about $60 \sim 80 \text{ mJ} \cdot \text{mol}^{-1} \cdot \text{K}^{-2}$ (smaller than those of typical rare-earth heavy-fermion systems but the largest among all Mn-based compounds) [He, Miller, Aronson, *Inorg. Chem.* **53**, 9115–9121 (2014), Li *et al.*, *Phys. Rev. B* **104**, 134305 (2021)] and power-law behaviors in both C_p/T [He, Miller, Aronson, *Inorg. Chem.* **53**, 9115–9121 (2014)] and magnetic susceptibility below 10 K (see Fig. 2(c) in the manuscript), altogether signifying enhanced electron correlations via heavy fermion mass and also critical spin dynamics in the proximity of ferromagnetic ordering. There have been observations of heavy-fermion metals with ferromagnetism in systems like CeRh_6Ge_4 . Curie temperature of the compound is $T_C = 2.5$ K, it shows a crossover from the high- T antiferromagnetic (AF) Curie-Weiss behavior to the suppression of the inverse susceptibility below 30 K, similar to our system [Matsuoka *et al.*, *J. Phys. Soc. Japn.* **84**, 073704 (2015)]. The presence of heavy-fermion flat bands and other dispersive bands close to the Fermi level in CeRh_6Ge_4 has been directly confirmed by a recent ARPES measurement [Wu *et al.*, *Phys. Rev. Lett.* **126**, 216406 (2021)], although potential spin splitting of the flat bands below T_C is yet to be confirmed. Since our Mn-kagome compound hosts both heavy kagome-induced flat bands and light dispersive bands with some hybridization between them, we suggest that heavy-fermion-like ferromagnetism as observed in CeRh_6Ge_4 can be relevant to the ferromagnetic fluctuations in our system. Note that, there has been a growing interest in the universality of Kondo-like physics not only in rare-earth compounds but also in transition-metal-based kagome systems and iron superconductors, and even in twisted layered systems, all hosting the

Figure 2: Paramagnetic DFT+RISB bands with (a) $J_H = 1.0$ and (b) 1.5 eV, where U is set to be 15 eV. Note that all the bands in the energy range are Mn d -orbital-originated. (c) Ferromagnetic solution at $(U, J_H) = (15, 1.8)$ eV. Blue and red curves depict spin up and down bands respectively.

coexistence of flat and dispersive bands near the Fermi level [Paschen and Si, *Nat. Rev. Phys.* **3**, 9–26 (2021), Checkelsky *et al.*, arXiv:2312.10659].

On the Reviewer’s suggestion of enhancing spin polarization and simulating the magnetoresistance for a comparison with experimental results, there are two practical difficulties; *i*) due to the computational cost issue, our DMFT calculation cannot approach low-temperature regime below 50 K where the magnetic fluctuation is expected to manifest, and *ii*) magnetic constraint to enforce spin polarization has not been implemented with the DMFT formalism that we are employing in this study, which is necessary to check the Review’s suggestion because all of our DMFT calculations starting from FM and AFM initial states converge to paramagnetic solutions. But we believe that, our approaches as described above address the Reviewer’s concern in an alternative manner.

Changes to the manuscript : At the end of the **Flat Band Realized via Correlation-induced Orbital-dependent Band Energy Renormalization** section we added a paragraph on our new DFT+RISB calculation results and the observed ferromagnetic phase. Also in the new **Discussion** section we inserted a new paragraph on the similarity between our system and ferromagnetic heavy-fermion systems.

Criticism-2: *If as the authors claim, the B_g orbitals are selectively responsible for the enhancement in the ferromagnetism, do they observe direction dependent change in magneto resistance. In the manuscript they have presented magnetoresistance with magnetic field in the Kagome plane (no specific direction has been measured). Does the magnetoresistance have angular dependence in the ab plane or to the c axis? Are the ferromagnetic fluctuations in the Kagome plane or out of plane? Could the authors also please mention the direction of applied magnetic field in the magnetic susceptibility and MH measurements? In addition, do they observe anisotropic behaviour in the magnetism at low temperatures in the in-plane and out-of-plane directions?*

Response: In accordance with the Referee's recommendations, we undertook a comprehensive analysis comparing the magnetic susceptibility, magnetization, and MR between the in-plane and out-of-plane orientations. This inspection revealed several noteworthy features:

- Distinct magnetic correlations develop in several temperature intervals for in-plane and out-of-plane magnetic susceptibilities.
- The magnetic behavior becomes more isotropic as the temperature is lowered from 145 K, deduced from the ratio of χ_{ab}/χ_c in the inset of new Fig. 2(b).
- The internal mean field $H_{MF}(T)$ derived from the in-plane magnetization is roughly twice that obtained from the out-of-plane $M(H)$. The directional dependence of MR resembles that of $H_{MF}(T)$. These intriguing variations in magnetic characteristics between the two orientations are compatible with a multi-energy scale.

Based on our new dataset, we conclude that a weakly XY-like magnetism becomes more isotropic on cooling through 145 K and that the ferromagnetic fluctuations are notably more pronounced for the Kagome plane than the out-of-plane directions. The MR is a consequence of ferromagnetic instabilities.

Changes to the manuscript : Fig. 2 and Fig. 6 were updated in the revised manuscript for the presentation of new magnetic susceptibility and MR data, and relevant discussions were

also updated in the revised manuscript.

Criticism-3: *Describing Fig. 4(g) the authors state that that orbital-dependent mass enhancement and shift of $\text{Re}\Sigma(0)$ are strongest in the B_g orbitals. This is clearly not the case in Fig. 4(g) where B_g and also the authors note later in the manuscript that the mass enhancement of A_g^{in} orbital is also comparable to that of B_g . These statements need further clarification and consistency. Also, the authors do not show independently how the energy renormalisation behaves for A_g^{in} orbitals w.r.t. U , and a comparison of energy renormalisation vs temperature between B_g and A_g^{in} orbital. A substantial difference may help elucidate their claim of dominant role of B_g orbital, despite its comparable mass enhancement as A_g^{in} orbital. Also, the authors do not show the orbital susceptibility change as a function of temperature for the A_g^{in} orbital in Fig. 5(b).*

Response: There are two comments here, so we answer them one by one below.

- On our confusing statement on the mass enhancement of B_g orbitals. We thank the Reviewer for pointing out this. Indeed, the amount of mass enhancement of the A_g^{in} orbital, as depicted in Fig. 4(g), is comparable to that of the B_g one. On the other hand, the orbital differentiation between the B_g and A_g^{in} is clearer from the energy renormalization, namely from the real part of the zero-frequency self-energy. In our Fig. 4(h) we show the relative mass renormalization of A_g^{in} and A_g^{out} orbitals with respect to that of B_g , *i.e.* $\text{Re} \left[\Sigma^{A_g^{\text{in}}} - \Sigma^{B_g} \right] (\omega = 0)$ and $\text{Re} \left[\Sigma^{A_g^{\text{out}}} - \Sigma^{B_g} \right] (0)$, respectively. To corroborate this, as the Reviewer suggested, we added a new panel (j) on the temperature evolution of the relative energy renormalizations in Fig. 4 in the revised manuscript (see Fig. 3 in this letter). This temperature dependence also shows that the renormalized energy difference between the B_g and A_g^{in} orbitals is enhanced as the temperature is lowered. Note also that the energy shift of the B_g band upon electron correlations is evident from Fig. 3 and 4 in the manuscript, while the peak of the A_g^{in} band close to the Fermi level in the DFT results (see Fig. 3(c) and (d) in the manuscript) is almost untouched in the DMFT result (Fig. 3(g) and (h)).

Changes to the manuscript: In the revised manuscript we have corrected the phrase ‘orbital-dependent mass enhancement and shift of $\text{Re}\Sigma(0)$ are strongest in the B_g orbitals’ and updated in the main text. And we added a new panel (j) in Fig. 4, showing the temperature-dependence of the self-energy differences.

- On the orbital susceptibility data plotted in Fig. 5(b) of the manuscript. For the compu-

Figure 3: Energy renormalization: $\text{Re}(\Sigma_{B_g} - \Sigma_{A_g^{\text{out}}})$ (green) and $\text{Re}(\Sigma_{B_g} - \Sigma_{A_g^{\text{in}}})$ (red) versus temperature at zero frequency for $(U, J_H) = (8, 0.8)$ eV. This plot is used as the new panel (j) of the new Fig. 4 in the revised manuscript.

tation of the orbital susceptibility, in this work we followed the formulation employed in X. Deng, *et. al.*, Nat. Commun. **10**, 2721 (2019), *i.e.* $\chi_{\text{orb}} = \int_0^\beta \langle \Delta N_{\text{orb}}(\tau) \Delta N_{\text{orb}} \rangle d\tau - \beta \langle \Delta N_{\text{orb}} \rangle^2$, where ΔN_{orb} is defined as the occupation difference between two orbitals (or the difference in average occupations of the two groups of orbitals) within the chosen orbital multiplet. In other words, we are interested in charge fluctuations within an orbital sector at our choice, and contrasting behaviors of χ_{orb} depending on the choice of orbital sectors can be a signature of orbital differentiation between them. Indeed, Fig. 5(b) shows a clear difference between B_g and A_g^{out} sectors, namely, the charge fluctuation within the B_g orbitals become sharply enhanced as the temperature is lowered while it is almost constant within the A_g^{out} sector. This sharp enhancement of χ_{orb} in the B_g sector is a signature of electron localization and the tendency toward the orbital-selective Mott transition therein.

Since our choice of χ_{orb} can be only defined in the presence of multiple orbitals, this cannot be computed only for the single A_g^{in} orbital. It is possible to compute χ_{orb} between different orbital sectors, for example between B_g and A_g^{out} , but we believe that the current presentation as depicted in Fig. 5(b) suffices to deliver our message on the orbital differentiation and the tendency toward the orbital-selective Mott transition.

Changes to the manuscript : In the **Orbital Decoupling and Bad Metallic Phase by Hund's Coupling** section of the revised manuscript, relevant discussion in the fourth paragraph was updated.

Criticism-4: *It has been concluded in the manuscript that the resistivity upturn at low temperatures is a result of enhanced electronic scattering, associated with development of ferromagnetism in the system. With the development of ferromagnetism, one would expect increase in spin polarisation and hence smaller electron-electron scattering and lower resistivity. Could the authors explain this observation?*

Response: We comment that, despite we observed experimental signatures of ferromagnetic fluctuations below $T = 10$ K (power-law behavior of magnetic susceptibility, enhancement of the size of the Weiss molecular field, nuclear spin-spin relaxation rate from NMR), no signature of ferromagnetic ordering has been observed in magnetic susceptibility and specific heat data [He, Miller, Aronson, *Inorg. Chem* **53**, 9115–9121 (2014)] of our system down to $T = 2$ K. We believe the small hysteresis loop is an instrumental artifact. We deleted it in the new Fig. 2 in the revised manuscript. Here better to emphasize that our system is on the verge of ferromagnetic instabilities, we experimentally observe the development of ferromagnetic correlations (not ferromagnetic ordering). Much like Kondo scattering, weakly fluctuating magnetic moments can offer an efficient scattering channels to itinerant charge carriers. Exerting magnetic fields should reduce this ferromagnetic fluctuation and the resulting electron and spin scattering, yield the negative magnetoresistance, as the Reviewer suggested.

Changes to the manuscript : In the revised manuscript we have made more clear that no long-range magnetism is observed by removing the hysteresis data from the new Fig. 2.

Further comments:

Criticism: *It might be helpful for readers, if the authors can include in Figure 1, schematically, the five local d orbital orientations w.r.t. the Kagome planes in addition to the coloured arrows used to represent the axes at present.*

Response: We have accepted the reviewer suggestion and modified the Fig. 1 accordingly. Please check Figure 3 below in this rebuttal letter.

Changes to the manuscript : A new panel (c) in the Fig. 1, showing the five Mn d -orbitals with respect to the Mn kagome plane, was inserted in the revised manuscript.

Criticism: *The authors mention, ‘Below 1500 cm^{-1} , where the intra- and interband transitions are roughly divided, a spectral weight transfer from low- to high-frequency regimes occurs as the temperature is lowered, resulting in the sharpening of the Drude peak.’ It appears however, as*

Figure 4: Schematic of five Mn d orbitals shown with respect to Kagome plane with crystal and local axes defined and indicated in colored arrows.

if spectral weight transfer is transferred from high- to low-frequency regimes as the temperature is lowered, which would indicate the gradual band movement towards the Fermi level.

Response: We thank reviewer for pointing out this inadvertent mistake. We have corrected this sentence and updated in the revised manuscript.

Reviewer #2 (Remarks to the Author):

Response: We thank co-reviewer for reviewing our manuscript and providing useful comments.

Reviewer #3 (Remarks to the Author):

In this work, the authors employed a collection of experimental and theoretical techniques to characterize the electronic properties of $Sc_3Mn_3Al_7Si_5$ (SMAS), which contains a kagome substructure of Mn atoms. Experimentally, the authors observed the presence of ferromagnetic fluctuations at low temperatures in spite of a negative Curie-Weiss temperature. This was inferred from a small magnetization hysteresis and ^{27}Al NMR. The effective magnetic moment found was also substantially smaller than the spin-only value expected for a Mn^{3+} ion, which indicated itinerant magnetism. Additionally, the authors observed a significant negative magne-

toresistance and a slight suppression of the Drude peak in the optical conductivity, which were interpreted as being consistent an enhancement in electron scattering due to the ferromagnetic fluctuations. On the theory front, the authors first identified a set of relatively flat bands from the DFT band structure and confirmed their Wannier functions resemble the shape expected for the flat bands arising from a kagome structure. However, these flat bands are energetically quite far away from the Fermi level. The authors then showed that electronic correlations could bring these flat bands closer to the Fermi level as the system becomes a correlated metal which is incipient to an orbital selective Mott phase, in the sense that incoherent spectral weight associated with the upper and lower Hubbard bands have developed but a coherent spectral peak remained. Further evidence was provided to support the orbital dependence on the correlation effect.

Overall, this manuscript presented a collection of interesting experimental phenomena observed at low temperature in SMAS and combined that with a theoretical study on the expected effects of electronic correlation on this compound.

Response: We are thankful to Reviewer for finding our experimental and theoretical results interesting.

While these results are certainly interesting in their own rights, this Reviewer judged that the two aspects of the study, namely experimental and theoretical results, were not truly tied together. More specifically, the experimental results all correspond to (indirect) evidence for the presence of significant ferromagnetic fluctuations at low temperatures, whereas the theoretical results indicated a flattening and shifting of the B_g -bands as a set of flat bands at the Fermi level. The relation between the theory and the experiments then rest upon the idea of flat-band ferromagnetism, i.e., the assumption that electrons occupying a flat band would tend to develop ferromagnetic order. While this is a believable explanation of the experimental results, it is also a rather indirect conclusion and there are many subtleties one should be concerned with in pursuing this interpretation.

For instance, as the authors noted the kagome flat bands arise from geometrical frustration. It is unclear to the present Reviewer if one could immediately generalize the existing results on flat-band ferromagnetism to such a setup involving nontrivial wave functions. The presence of a large number of additional bands also raise question on whether or not one can immediately expect a simple relation between the correlated flat bands and ferromagnetic fluctuation.

Also, due to computational cost the DMFT studies were restricted to temperatures which were one order of magnitude higher than that relevant for the experimentally interesting observations. Understandably, one could only draw general conclusions on the trend of, for instance, the

development of orbital-selective correlation etc. In the opinion of the Reviewer, however, it would be risky to take these general trends as support for an experimental proof of observing correlation-induced flat bands near the Fermi energy.

All in all, while this manuscript certainly highlight SMAS as a potentially interesting material which warrants further investigations, to meet the high standards of Nature Communications, the Reviewer believes a much stronger link between the theoretical and experimental studies presented around SMAS would need to be established.

Response: We agree with the Reviewer’s comment on the weak link between the theoretical and experimental results of our previous manuscript, and this issue was also pointed out by the Reviewer 1 (see the Criticism 1 by the Reviewer 1). Also, we thank the Reviewer for his/her understanding of the computational cost issue of the currently employed DMFT method in approaching the zero-temperature limit. It should be noted that direct confirmation of the theory-predicted flat bands via experimental probes such as ARPES or quantum interference from scanning tunneling microscopy is a challenging task in our system because of the difficulty in sample cleaving and polishing.

To approach the zero-temperature limit on the calculation side, which is beyond the capability of our DMFT method employing the continuous-time quantum Monte Carlo impurity solver, we employed the rotationally-invariant Slave-boson (RISB) mean-field theory in combination with DFT (DFT+RISB). Our DFT+RISB result reproduces basically the same results with our DMFT ones; strong band flattening and energy renormalization of the B_g -flat bands toward the Fermi level. Furthermore, from the DFT+RISB results it is found that the ferromagnetic instability emerges only when the B_g -flat bands are very close to the Fermi level (see Fig. 2 for the ferromagnetic spin splitting from the DFT+RISB results), supporting a link between our calculations and experimental observations.

Also, from the literature search, we found other examples of ferromagnetic instabilities in systems with flat and multiple dispersive bands coexisting in the vicinity of the Fermi level; for example, ferromagnetism in a heavy-fermion metal CeRh_6Ge_4 [Matsuoka *et al.*, *J. Phys. Soc. Japn.* **84**, 073704 (2015)]. Note that our system shares some characteristics of heavy-fermion systems close to the critical area, *i.e.* a rather large Sommerfeld coefficient of about $60 \sim 80 \text{ mJ} \cdot \text{mol}^{-1} \cdot \text{K}^{-2}$ (smaller than those of typical rare-earth heavy-fermion systems but the largest among all Mn-based compounds) [He, Miller, Aronson, *Inorg. Chem.* **53**, 9115–9121 (2014), Li *et al.*, *Phys. Rev. B* **104**, 134305 (2021)] and power-law behaviors in both C_p/T [He, Miller, Aronson, *Inorg. Chem.* **53**, 9115–9121 (2014)] and magnetic susceptibility below 10 K (see Fig. 2(c) in the manuscript), altogether signifying enhanced electron correlations via

heavy-fermion mass and also critical spin dynamics in the proximity of ferromagnetic ordering. Hence we believe that reports of ferromagnetic heavy-fermion metals can be an indirect bridge between our theoretical and experimental findings.

For more details please refer to our answer to Criticism 1 of the first reviewer.

Changes to the manuscript : At the end of the **Flat Band Realized via Correlation-induced Orbital-dependent Band Energy Renormalization** section we added a paragraph on our new DFT+RISB calculation results and the observed ferromagnetic phase. Also in the new **Discussion** section we inserted a new paragraph on the similarity between our system and ferromagnetic heavy-fermion systems.

Reviewer #4 (Remarks to the Author):

In this manuscript the authors explore the electronic properties of the Kagome materials $Sc_3Mn_3Si_5Al_7$ through a combination of susceptibility, NMR, transport and optics measurements as well as band calculations.

There has been a lot of recent interest in metallic Kagome systems. Unlike the previous focus on the magnetic frustration of already localized spins such as what is found in the insulating Kagome materials Herbertsmithite, the interest in metallic Kagome systems has focused on the electronic structure and the formation of flat bands due to the Kagome structure. The flat bands in Kagome systems stem from localization of charge due to destructive interference caused by the geometry of the Kagome lattice. This localization and formation of flat bands is an important precursor for several interesting topological and non-topological phases. Therefore the study of new flat band metallic Kagome materials is a relevant and important endeavor.

Response: We thank Reviewer for appreciating and finding our work relevant.

Criticism-1: *The authors have a comprehensive study of DFT and DMFT calculations on the formation of flat bands and the orbital contributions that form these flat bands.*

The authors show some magnetic susceptibility data which is similar to what has been previously presented and is only tenuous evidence at best for ferromagnetic order.

The authors claim there is a ferromagnetic instability below 30 K, however their Curie-Weiss fit shows a large antiferromagnetic coupling. The authors should comment on the relative energy scale of the ferromagnetic instability at low temperature compared to the mean-field antiferromagnetic coupling at high temperature.

Response: We thank the Reviewer for asking the important question on energy scales of

magnetism in low- and high-temperature regimes. As the referee commented, the presence of the low-temperature ferromagnetic instability is somewhat hard to understand due to the presence of the large negative Weiss temperature. We want to answer this comment in a few different directions. Please also refer to our answers to Criticism 1 of Reviewer 1 and the comments from Reviewer 3.

- On the (indirect) evidence of low-temperature ferromagnetism in our system.
 - We checked that, in our DMFT calculations, no magnetic order is favored down to $T = 116$ K. On the other hand, our first-principles rotationally-invariant Slave boson calculation (DFT+RISB) [Lanata *et al.*, *Phys. Rev. X* **5**, 011008 (2015), Lanata *et al.*, *Phys. Rev. B* **87**, 045122 (2022)], which captures the effects of electron correlations of the metallic phase at the zero-temperature limit, shows a subtle but finite ferromagnetic order when the B_g -induced flat bands are very close to the Fermi level (see Fig. 2).
 - Experimentally, we have added the directional dependence of magnetic susceptibility, magnetization, and MR. As detailed in the reply to the Reviewer 1, our new data demonstrate that the high-temperature magnetism, dictated by frustrated XY-like antiferromagnetic correlations, experiences a multi-stage thermal evolution. This behavior cannot be captured within a single orbital picture or dominant AFM interactions with weak FM interactions.
- Other similar systems. We have found that several other ferromagnetic heavy-fermion compound, like CeRh_6Ge_4 [Matsuoka *et al.*, *J. Phys. Soc. Jpn.* **84**, 073704 (2015)], showing a crossover from the high-temperature antiferromagnetic behavior (with a large Weiss temperature from the magnetic susceptibility) to the low-temperature ferromagnetism. In addition, the presence of flat heavy-fermion bands and other dispersive bands in the vicinity of the Fermi level has been shown, like our Mn-kagome system [Wu *et al.*, *Phys. Rev. Lett.* **126**, 216406 (2021)].
- On the microscopic mechanism of the crossover. We believe that, while the Mn moments show dominantly local-moment behavior in the high-temperature regime, the local moment degree of freedom tends to couple to the low-energy electronic degrees of freedom to form weakly-dispersive bands close to the Fermi level. This behavior has been observed in correlated metals close to the Mott transition, or in rare-earth heavy-fermion systems [Rozenberg *et al.*, *Phys. Rev. Lett.* **75**, 105 (1995), Georges *et al.*, *Rev. Mod. Phys.* **68**,

13 (1996)]. It is interesting to note that there has been a growing interest in the universality between heavy-fermion systems and 3d-transition metal compounds with orbital-selective electron correlations such as iron-based superconductors [Paschen and Si, *Nat. Rev. Phys.* **3**, 9–26 (2021), Checkelsky *et al.*, arXiv:2312.10659]. Then, it is likely that the high-temperature Curie-Weiss behavior in our system is overwhelmed by the low-temperature formation of the flat-band Fermi surface and the ferromagnetic (FM) responses induced by the flat bands.

- On the energy scale of the low-temperature ferromagnetic fluctuations. Our mean-field analysis of the M - H curves yields the internal field of 0.1 T at 2 K, which is more than three orders of magnitude smaller than the high-temperature Curie-Weiss temperature and the typical internal field of ferromagnets (an order of 100 T). Theoretical studies on metallic ferromagnetism induced by the flat bands report that, while many aspects of the flat-band-induced ferromagnetism are qualitatively consistent with the Stoner mean-field description, the ferromagnetic energy scale (namely the Curie temperature, which should be proportional to the product of the on-site Coulomb U and the magnetization) is strongly suppressed by the subtle cooperation between the electron kinetic energy and the electron correlations [Tasaki, *Prog. Theor. Phys.* **99**, 489–548 (1998), Brando *et al.*, *Rev. Mod. Phys.* **88**, 025006 (2016)]. At this point, there seems to be no clear explanation on the energy scale of the ferromagnetic fluctuation in our system, especially in the presence of multiple orbitals and other dispersive bands. We believe that this is definitely an intriguing question to pursue in our future study.

Changes to the manuscript : At the end of the **Flat Band Realized via Correlation-induced Orbital-dependent Band Energy Renormalization** section we added a paragraph on our new DFT+RISB calculation results and the observed ferromagnetic phase. Also in the new **Discussion** section we inserted a new paragraph on the similarity between our system and ferromagnetic heavy-fermion systems. Lastly, in the new Fig. 2 and relevant parts in the revised manuscript, discussion on the magnetic energy scales in the low-temperature regime was improved.

Criticism-2: *The authors present as well measurements of the magnetoresistance and optical conductivity. For those measurements they comment it is consistent with their calculations but provide little more conclusions to draw from it.*

Response: We thank the Reviewer for pointing out this. In the revised manuscript we have more detailed discussion on the temperature dependence as well as on the direction-dependence

of the magnetoresistance data with respect to the orientation of the B -field. Please understand that a rigorous discussion on the detailed character of the MR signal based on our electronic structure calculations is challenging due to the multiorbital nature of our system. At this point we can only say that the negative MR is correlated with the development of FM internal fields.

Specific comments on the paper

Criticism-3: *In page 2 second to last paragraph, the authors claim that the magnetic susceptibility is a sign of weak ferromagnetism. ‘Weak ferromagnetism’ is a feature of the dzyaloshinskii-moriya interaction resulting in a net moment within an otherwise antiferromagnetic material. If the material does not have long range antiferromagnetic order, this would not explain a magnetic hysteresis through weak ferromagnetism. In fact this hysteresis plot is not very convincing, is it sample independent? What is its temperature dependence?*

Response: We thank the Reviewer for spotting this issue. As the Reviewer correctly pointed out, the terminology "weak ferromagnetism" is misleading. We have checked the existence of weak hysteresis loop and confirmed that it is due to an offset of VSM during the up and down sweeps. Accordingly, we deleted it. We stress that all magnetic quantities indicate only the development of weak internal fields (not long-range magnetic ordering).

Criticism-4: *Fig. 2a, and 2b are the same as has been previously reported with similar results.*

Response: Following the Reviewer comment, we have remeasured the directional dependence of magnetic susceptibilities, performed further analysis, and updated the Fig. 2 in our revised manuscript. As for the resistivity, we think it is important to present its figure in the main text to avoid sample dependence issue. In the revised version, we explicitly state the consistency of our data with the previous works.

Criticism-5: *Fig. 2b, what is the χ_0 used in the Curie Weiss fit and how was it found? This information should be in the Supplementary materials. How does this χ_0 compare in a Sommerfeld-Wilson ratio?*

Response: Our compound features orbital-selective correlations. In this regime, it is not straightforward to estimate χ_0 , which comprises of diamagnetic and Pauli-like paramagnetic contributions. In the coexistence weak local moments and itinerant carriers, estimation of the Sommerfeld-Wilson ratio from χ_0 might be not well justified. As such, we present CW analysis without χ_0 subtraction and do not pursue the further analysis in the revised manuscript.

Figure 5: Projected band structures display the orbital contribution of Mn, Sc $-d$ and Si, Al $-p$ orbitals from nonmagnetic DFT calculation. In addition, momentum and frequency dependent spectral function from DMFT calculation for five Mn d orbitals is shown.

Criticism-6: *Fig. 2c. Is there really much evidence for a second power law scaling in the range from 10-20K. I point out that any smooth curve will appear linear over a small enough range, which is what appears to be happening in this case.*

Response: We thank the Reviewer for bringing our attention to this issue. We have remeasured the in- and out-of-plane magnetic susceptibilities (see the new Figure 2(c) in the revised manuscript). We could identify the power-law behavior above 20 K. However, as the Reviewer pointed out, there is a crossover between 7-20 K and eventually, another power-law behavior appears below 7 K. Nonetheless, the power-law behavior at low temperatures should be regarded as an approximation, enabling the discrimination of orientation and thermal characteristics of the magnetic behavior. We provide a more careful description of the magnetic susceptibility in the revised manuscript.

Criticism-7: *Fig. 2e, Could this hysteresis be from any Mn impurity on the sample. What is its temperature and sample dependence.*

Response: We thank the Reviewer for raising this issue. As replied above, the hysteresis is not intrinsic to the studied compound, and the plot was removed in the manuscript.

Criticism-8: *Fig. 3, it would be useful if the Authors included a picture of the Brillouin zone showing the M, K, Γ , A, H and L points.*

Response: We thank Reviewer for this useful suggestion. In the new Fig. 3, we have added a schematic of the Brillouin zone.

Criticism-9: *Fig. 3 a-c and e-f show the relative contribution of three different Mn-orbital irreps. However this does not show whether the flat band is predominantly formed from contributions on the Kagome lattice. The authors should include a similar figure (perhaps in the Supplementary materials) showing the relative contribution of each atom to the bands.*

Response: We thank Reviewer for this suggestion. We include a new figure in the supplementary, showing the orbital contribution of individual Mn d orbitals, Sc d , Al and Si p orbitals. Figure 5 clearly shows that flat bands predominantly originate from the kagome lattice, formed by Mn atoms. It also confirms that flat bands carry orbital character of d_{xz} and d_{yz} .

Criticism-10: *Fig. 5c what is the peak at approximately 200 K in the spin susceptibility? Is this a calculation artifact or a real effect?*

Response: The kink at 150 K in Fig. 5c is not a mere calculation artifact. Rather it appears to be a real, likely the turning point from where spin susceptibility follow decreasing trend. We confirm it from our new susceptibility vs temperature plot, as shown in Fig. 4. Earlier, spin susceptibility was computed at each temperature for $M=96\times 10^9$ Monte Carlo steps. Now we increase number of Monte Carlo steps to $M=752\times 10^9$ and recomputed spin susceptibility. The new plot looks almost similar to what we reported earlier in our manuscript. Since low temperature calculations incur much computational cost, we restrict ourselves not to go below 120 K. At this point, we further comment that the peak-like structure associated with spin susceptibility cannot be confirmed whether it is related to the Hund's metal physics. Note that orbital and spin susceptibilities versus energy (temperature) curves of Hund's metal feature such peak structure, where orbital degrees of freedom quenches prior to the spin one (X. Chen *et. al.*, Nat. Commun. **11**, 3076 (2020)). Since we cannot lower the temperature below 120 K, such feature cannot be ascertained in the current study.

Criticism-11: *On the section ORBITAL DECOUPLING AND BAD METALLIC PHASE*

Figure 6: DC spin susceptibility versus temperature

BY HUND'S COUPLING, the authors say that with a Hund's coupling of 0.8 eV, there is a moment size of 0.9 μ_B at 300 K. Does the increasing orbital moment at lower temperatures imply a larger moment with a larger orbital contribution at lower temperatures? What is the expected moment at 200 K? If the moment is changing between 300 and 200 K, does that fit for the Curie-Weiss fit?

Response: The size of local spin magnetic moment of Mn is 0.9 μ_B at 300 K, which remains the same at 200 K (change being smaller than 0.01 μ_B). The value of spin moment agrees well with experimental moment of 0.86 μ_B /Mn, obtained from our Curie-Weiss fit. On the other hand, gradual increase in χ_{orb} for B_g orbitals is associated with orbital decoupling and quenching of one of the orbital degrees of freedom within B_g sector.

Criticism-12: *Fig. 6, I point out that the increase in the conductivity at 2 K (Fig. 2a) is approximately 4%, the same quantity that is found in the magnetoresistance. This means that at 9 T, the low temperature upturn in resistivity is likely suppressed.*

Response: Following the Reviewer's advice, we have added field- and temperature dependence of the electrical resistivity in the inset of Fig. 2a. Our new data clearly demonstrate a systematic suppression of the low-T resistivity with increasing field towards 9 T.

Criticism-13: *In the discussion, the authors state that the hysteresis at 2 K is a sign of ferromagnetic fluctuation. I disagree, only an ordered state would have non-zero magnetization at zero field and show that kind of curve. Ferromagnetic fluctuations at zero field will average out to a zero net moment at zero field. Another way to think about this is that in order to have a net ferromagnetic moment at zero field the material needs to have passed a point where the*

time correlation has diverged, this will only happen after a phase transition has passed.

Response: We apologize for wrong assertion based on the non-intrinsic effects. Given all other characterizations, our system precludes long-range ordering. Once again, we thank the Reviewer for helpful comments.

Comments on the supplementary materials

Section IV the authors mention Korringa law for temperature dependence. They should add a citation explaining what this is.

Response: We have added proper citations for broader readers.

Section IV. the authors include a number 44.8833 ± 0.24882 . No number should have more than 1-2 digits of uncertainty presented.

Response: We have accepted reviewer's suggestion and keep the number up to second decimal place.

REVIEWER COMMENTS

Reviewer #1 (Remarks to the Author):

The authors have improved and corrected parts of the manuscript considering the suggestions and have responded effectively to the comments made by the reviewers. I have no further concerns with this manuscript. The authors have provided further arguments to the claim of ferromagnetic instabilities and I believe from the scope of the present of the work, the present manuscript would stimulate further works in this field. I have no further concerns about the manuscript.

Reviewer #2 (Remarks to the Author):

Reviewer #3 (Remarks to the Author):

I have reviewed the previous round of referee reports, the authors' replies, the revised manuscript, and the supplementary information. The authors have made various corrections and revisions to their descriptions of the experimental results in response to the concerns raised by the other referees. The major concern raised by the present referee, namely the disconnect between the experimental and theoretical results, has also been addressed.

Specifically, the authors have included more discussions and clarifications on the DFT+U calculation, which indicates that the DFT+U calculation leads to opposite behavior on the shifting of the B_g bands and also favors antiferromagnetic order unless an unrealistic large value of U is used. Additionally, the authors have performed additional calculations with DFT + rotationally invariant slave bosons, which serves as a complementary calculation approach to DMFT from the zero-temperature limit. The DFT+RISB results are in broad agreement with the DMFT results, and in particular, help confirm the initially proposed picture of flat-band ferromagnetism as a possible explanation of the experimentally observed ferromagnetic fluctuations.

Based on these new results, I am convinced that the authors have strengthened the connections between their experimental and theoretical results, and have demonstrated the correlation-driven emergence of flat bands as a reasonable explanation of the observed enhancement of ferromagnetic fluctuations in SMAS. Therefore, I recommend publishing the manuscript in Nature Communications.

As a small suggestion, I would also encourage the authors to consider including the observation of ferromagnetic fluctuations as part of the title.

Reviewer #4 (Remarks to the Author):

In this resubmission the authors have made several improvements to the paper and addressed key concerns.

The addition of directional dependence of the magnetic susceptibility and magnetoresistance are illuminating. Most importantly, the authors were able to determine that the unusual feature found in the previous M vs H plot was in fact a measurement artifact. Subsequently the authors no longer used this data for their conclusions, which strengthens their argument.

There is however a significant error in the analysis and interpretation of the magnetization data shown in Figure 2 parts d, e and f.

The data in parts d and e of the figure are fit to a function consisting of $M(H,T) = NJg\mu_B B(J, \gamma) + \chi(T)H$, where J is the total angular moment, g is the g-factor, and $\gamma = Jg\mu_B(H + H_{MF})/kBT$. In this formulation, the authors claim that H_{MF} a Weiss molecular field. The problem with this formulation, is that the Weiss molecular field is an effective field that is a function of the current magnetization, usually written

$$H_{MF} = \lambda M.$$

Here λ is the constant related to the strength of the coupling. And M is the current magnetization.

In the paper here the authors use a single static field for the entire magnetization curve, where the range of values of the magnetization range all the way to zero, where there should be no Weiss molecular field.

In fact when looking closely at the fits for the curves in Fig. 2 d and e, it is clear that the fit curve does not go to zero, while the data appear to do so. For a material that is not ferromagnetically ordered, one expects the magnetization to be zero at zero field.

Specific comments

Regarding Fig. 1c, the authors should know that there is always a χ_0 contribution to the magnetic susceptibility. The most important contribution for a metal is likely the Pauli paramagnetism contribution.

This contribution is proportionally most important at high temperature where the other contributions to the susceptibility are smallest. Because of this, a fit to a power law in the high temperature regime, without subtracting a χ_0 term is not very meaningful. In fact, the fit to the power law in those regimes for the data presented do not appear to fit the data very well.

In addition, a power law fit with an exponent of -0.17 or -0.2 is inconsistent with presenting a Curie Weiss fit of the data (by definition a power law only with $\theta_{CW}=0$, and with an exponent of -1). In the first paragraph of page 3, the authors talk about this data and present both at the same time. Either the data at high temperature does not fit a power law, or a Curie-Weiss fit is not possible. In this reviewer's opinion high temperature power law fit should be omitted.

In Fig. 3 d and h. and Fig. 4, the authors give the density of states calculated, however in those plots the units are left as arbitrary units. It is certainly useful to readers to know the actual value of the density of states. I suggest changing the units to (states / unit cell / eV) as is done in (Liu et al, 2020) and (Kang et al, 2020). This allows comparing to other materials, and comparing to measured quantities such as Sommerfeld constant etc.

The magnetoresistance at low temperature, and high fields in a nearly ferromagnetic material should be proportional to $H^{-1/3}$ (Schindler and Rice 1967). It appears that the data at 2 K and fields above 5 T may have the same proportionality. Checking this relation would be a useful argument for this paper.

Minor comments

Fig. 1f, the authors have added a dashed line to the figure. They should add a description of what the dashed lines are in the figure caption.

Ref.

Liu et al, Nature Communications 11, 4002 (2020)

Kang et al, Nature Communications 11, 4004 (2020)

Schindler and Rice, Physical Review 164, 759 (1967)

Summary of changes made in the revised manuscript

List of changes made are as follows, where new texts inserted are colored in **Red** in the revised manuscript.

1. The title of the manuscript was changed to “*Emergence of flat bands and ferromagnetic fluctuations via orbital-selective electron correlations in Mn-based kagome metal*”, following the suggestion given by Reviewer #3 to mention the observation of ferromagnetic fluctuations in the title.
2. Plots of densities of states plots in the revised manuscript and supplementary materials are labeled with correct units (states / eV / formula unit).
3. New references mentioned by Reviewer #4 are cited in the revised manuscript.
4. Figures 2(c)-(f) have been updated by subtracting the χ_0 contribution and removing the power-law fits in the high-temperature range. Additionally, we have reanalyzed the magnetization using the Brillouin scaling model and plotted the amplitude parameter in Fig. 2(f).
5. Figure 6 has been updated to include the B^2 (or $B^{1.5}$) dependence for fields below 4.5 T, to clearly illustrate the deviation from conventional field dependence above 5 T, which is attributed to the presence of ferromagnetic correlations.

Response to REVIEWER COMMENTS

Reviewer #1 (Remarks to the Author):

The authors have improved and corrected parts of the manuscript considering the suggestions and have responded effectively to the comments made by the reviewers. I have no further concerns with this manuscript. The authors have provided further arguments to the claim of ferromagnetic instabilities and I believe from the scope of the present of the work, the present manuscript would stimulate further works in this field. I have no further concerns about the manuscript.

Response: We thank Reviewer for finding our revised manuscript and response to comments satisfactory and for appreciating our work.

Reviewer #2 (Remarks to the Author):

Response: We thank co-reviewer for reviewing our manuscript.

Reviewer #3 (Remarks to the Author):

I have reviewed the previous round of referee reports, the authors' replies, the revised manuscript, and the supplementary information. The authors have made various corrections and revisions to their descriptions of the experimental results in response to the concerns raised by the other referees. The major concern raised by the present referee, namely the disconnect between the experimental and theoretical results, has also been addressed.

Specifically, the authors have included more discussions and clarifications on the DFT+U calculation, which indicates that the DFT+U calculation leads to opposite behavior on the shifting of the B_g bands and also favors antiferromagnetic order unless an unrealistic large value of U is used. Additionally, the authors have performed additional calculations with DFT + rotationally invariant slave bosons, which serves as a complementary calculation approach to DMFT from the zero-temperature limit. The DFT+RISB results are in broad agreement with the DMFT results, and in particular, help confirm the initially proposed picture of flat-band ferromagnetism as a possible explanation of the experimentally observed ferromagnetic fluctuations.

Based on these new results, I am convinced that the authors have strengthened the connections between their experimental and theoretical results, and have demonstrated the correlation-driven

emergence of flat bands as a reasonable explanation of the observed enhancement of ferromagnetic fluctuations in SMAS. Therefore, I recommend publishing the manuscript in Nature Communications.

As a small suggestion, I would also encourage the authors to consider including the observation of ferromagnetic fluctuations as part of the title.

Response: We thank to Reviewer for being convinced with our revised manuscript and response to criticisms and recommending our manuscript for publication in Nature Communications. We also appreciate Reviewer's suggestion to include the observation of ferromagnetic fluctuations in the title of the manuscript and updated the title of the manuscript as suggested.

Reviewer #4 (Remarks to the Author):

In this resubmission the authors have made several improvements to the paper and addressed key concerns.

The addition of directional dependence of the magnetic susceptibility and magnetoresistance are illuminating. Most importantly, the authors were able to determine that the unusual feature found in the previous M vs H plot was in fact a measurement artifact. Subsequently the authors no longer used this data for their conclusions, which strengthens their argument.

There is however a significant error in the analysis and interpretation of the magnetization data shown in Figure 2 parts d, e and f.

The data in parts d and e of the figure are fit to a function consisting of $M(H,T) = NJg\mu_B B(J,y) + \chi(T)H$, where J is the total angular moment, g is the g -factor, and $y = Jg\mu_B(H + H_{MF})/k_B T$. In this formulation, the authors claim that H_{MF} a Weiss molecular field. The problem with this formulation, is that the Weiss molecular field is an effective field that is a function of the current magnetization, usually written

$$H_{MF} = \lambda M.$$

Here λ is the constant related to the strength of the coupling. And M is the current magnetization.

In the paper here the authors use a single static field for the entire magnetization curve, where the range of values of the magnetization range all the way to zero, where there should be no Weiss molecular field.

In fact when looking closely at the fits for the curves in Fig. 2 d and e, it is clear that the fit curve does not go to zero, while the data appear to do so. For a material that is not ferromagnetically ordered, one expects the magnetization to be zero at zero field.

Response: We fully agree with the Reviewer that our initial assumption was not well sup-

ported. Given that we are dealing only with moderate ferromagnetic correlations, deducing the Weiss molecular field is not appropriate. Consequently, we have adopted the Brillouin scaling model to more accurately describe the ferromagnetic behavior concurring with negative MR [P. Wagner et al., Phys. Rev. Lett. 81, 3980 (1998)]. It is important to note that this Brillouin scaling model has been previously successful in describing negative MR in the inhomogeneous paramagnetic phase of ferromagnetic manganites. For our case, the linear scaling model of the Brillouin function proved more effective in capturing ferromagnetic correlations compared to a quadratic scaling one. The adoption of this model enhances the accuracy and relevance of our analysis. Despite these changes, the core conclusion of our study remains unchanged. We have revised the corresponding paragraph to reflect these methodological changes and clarify our findings.

Specific comments

Regarding Fig. 1c, the authors should know that there is always a χ_0 contribution to the magnetic susceptibility. The most important contribution for a metal is likely the Pauli paramagnetism contribution.

This contribution is proportionally most important at high temperature where the other contributions to the susceptibility are smallest. Because of this, a fit to a power law in the high temperature regime, without subtracting a χ_0 term is not very meaningful. In fact, the fit to the power law in those regimes for the data presented do not appear to fit the data very well.

Response: (We believe that the Reviewer is referring to Fig. 2c in the manuscript.) The Reviewer correctly points out that χ_0 should be subtracted for an accurate Curie-Weiss fit. As previously mentioned, subtracting the constant χ_0 directly from our experimental data presents significant challenges. Consequently, we have estimated χ_0 using our density functional theory (DFT) calculations. According to these calculations, the correlation-induced flat-band feature diminishes in the high-temperature regime. This suggests that the simple DFT-computed density of states at the Fermi level, $D(\epsilon_F)$, becomes relevant in this context. Our nonmagnetic DFT calculations indicate $D(\epsilon_F) = 7.24$ states per [eV · formula unit (with 3 Mn)]. The value of χ_0 per mole can thus be calculated using the formula:

$$\chi_0 = \mu_0 N_A \mu_B^2 D(\epsilon_F),$$

where $N_A = 6.023 \times 10^{23}$ Avogadro's number, which yields $\chi_0 = 7.8 \times 10^{-5}$ emu · Oe⁻¹ · mol⁻¹ · Mn⁻¹. Accordingly, we have revised Fig. 2c.

In addition, a power law fit with an exponent of -0.17 or -0.2 is inconsistent with presenting

a Curie Weiss fit of the data (by definition a power law only with $\theta_{CW} = 0$, and with an exponent of -1). In the first paragraph of page 3, the authors talk about this data and present both at the same time. Either the data at high temperature does not fit a power law, or a Curie-Weiss fit is not possible. In this reviewer's opinion high temperature power law fit should be omitted.

Response: We are grateful to the reviewer for highlighting this issue. The reviewer accurately noted that simultaneously applying the Curie-Weiss and power-law fits can lead to conceptual inconsistencies, as the Curie-Weiss model is primarily suitable for systems with localized magnetic moments. Accordingly, we have removed the high-T power-law fits to prevent confusion among readers. In the revised manuscript, we clearly state this limitation of describing the intriguing magnetic behaviors in terms of either the Curie-Weiss or the power-law fits.

In Fig. 3 d and h. and Fig. 4, the authors give the density of states calculated, however in those plots the units are left as arbitrary units. It is certainly useful to readers to know the actual value of the density of states. I suggest changing the units to (states / unit cell / eV) as is done in (Liu et al, 2020) and (Kang et al, 2020). This allows comparing to other materials, and comparing to measured quantities such as Sommerfeld constant etc.

Response: We appreciate Reviewer's suggestion. New Fig. 3 and Fig. 4 include aforementioned changes suggested by the Reviewer.

The magnetoresistance at low temperature, and high fields in a nearly ferromagnetic material should be proportional to $H^{-1/3}$ (Schindler and Rice 1967). It appears that the data at 2 K and fields above 5 T may have the same proportionality. Checking this relation would be a useful argument for this paper.

Response: We appreciate the Reviewer for their insightful suggestions. As noted by the Reviewer, the MR(B) deviates from the conventional B^2 (or $B^{1.5}$) dependence at fields above 4.5 T. However, we were unable to observe the $B^{-1/3}$ dependence expected for nearly ferromagnetic materials. Considering the moderate ferromagnetic correlations in the studied compound, further high-field measurements are necessary to conclusively determine this dependence. At any rate, we have included this considerations and arguments in the revised manuscript.

Minor comments

Fig. 1f, the authors have added a dashed line to the figure. They should add a description of what the dashed lines are in the figure caption.

Response: (We believe that the Reviewer is referring to Fig. 2f in the manuscript.) We

appreciate the Reviewer's attention to the lacking description of the dashed lines in the figure caption of Fig. 2f. In the revised manuscript, we have explicitly stated that the thick line serves as a guide to the eye.

Ref.

Liu et al, Nature Communications 11, 4002 (2020).

Kang et al, Nature Communications 11, 4004 (2020).

Schindler and Rice, Physical Review 164, 759 (1967).

Response: We thank the Reviewer for pointing out these important references. Accordingly, in the revised manuscript we have cited them in appropriate positions.

REVIEWERS' COMMENTS

Reviewer #4 (Remarks to the Author):

The authors have further improved their manuscript. This reviewer finds this manuscript to present important and original results on a very unique and interesting material.

With the exception of correcting a typo described below, I find there are no further changes to be made and I recommend this manuscript to be published.

Typo:

In the new paragraph on Curie-Weiss analysis (paragraph starting with "Curie-Weiss (CW) analysis of $1/(\chi(T) - \chi_0)$ above 150 K"), there appears to be a typo in the naming of the resulting parameters. There are two results and both are named with a superscript c, likely one should have a superscript a or ab.

Response to REVIEWERS' COMMENTS

Reviewer #4 (Remarks to the Author):

The authors have further improved their manuscript. This reviewer finds this manuscript to present important and original results on a very unique and interesting material.

With the exception of correcting a typo described below, I find there are no further changes to be made and I recommend this manuscript to be published.

Typo:

In the new paragraph on Curie-Weiss analysis (paragraph starting with "Curie-Weiss (CW) analysis of $1/(\chi(T) - \chi_0)$ above 150 K"), there appears to be a typo in the naming of the resulting parameters. There are two results and both are named with a superscript c , likely one should have a superscript a or ab .

Response: We thank Reviewer for finding uniqueness and originality in our work and also for recommending the manuscript for publication.

We thank reviewer for careful reading and pointing out the typo. In the revised manuscript, we have corrected the typo.